# COCKPIT: A Practical Debugging Tool for the Training of Deep Neural Networks

**Frank Schneider**[*]
University of Tübingen
Maria-von-Linden-Straße 6
Tübingen, Germany
f.schneider@uni-tuebingen.de

**Felix Dangel**[*]
University of Tübingen
Maria-von-Linden-Straße 6
Tübingen, Germany
f.dangel@uni-tuebingen.de

**Philipp Hennig**
University of Tübingen &
MPI for Intelligent Systems
Tübingen, Germany
philipp.hennig@uni-tuebingen.de

## Abstract

When engineers train deep learning models, they are very much "flying blind". Commonly used methods for real-time training diagnostics, such as monitoring the train/test loss, are limited. Assessing a network's training process solely through these performance indicators is akin to debugging software without access to internal states through a debugger. To address this, we present COCKPIT, a collection of instruments that enable a closer look into the inner workings of a learning machine, and a more informative and meaningful status report for practitioners. It facilitates the identification of learning phases and failure modes, like ill-chosen hyperparameters. These instruments leverage novel higher-order information about the gradient distribution and curvature, which has only recently become efficiently accessible. We believe that such a debugging tool, which we open-source for PYTORCH, is a valuable help in troubleshooting the training process. By revealing new insights, it also more generally contributes to explainability and interpretability of deep nets.

## 1 Introduction and motivation

Deep learning represents a new programming paradigm: instead of deterministic programs, users design models and "simply" train them with data. In this metaphor, deep learning is a meta-programming form, where *coding* is replaced by *training*. Here, we ponder the question how we can provide more insight into this process by building a *debugger* specifically designed for deep learning.

Debuggers are crucial for traditional software development. When things fail, they provide access to the internal workings of the code, allowing a look "into the box". This is much more efficient than re-running the program with different inputs. And yet, deep learning is arguably closer to the latter. If the attempt to train a deep net fails, a machine learning engineer faces various options: Should they change the training hyperparameters (how?); the optimizer (to which one?); the model (how?); or just re-run with a different seed? Machine learning toolboxes provide scant help to guide these decisions.

Of course, traditional debuggers can be applied to deep learning. They will give access to every single weight of a neural net, or the individual pixels of its training data. But this rarely yields insights

---

[*]Equal contribution

35th Conference on Neural Information Processing Systems (NeurIPS 2021).

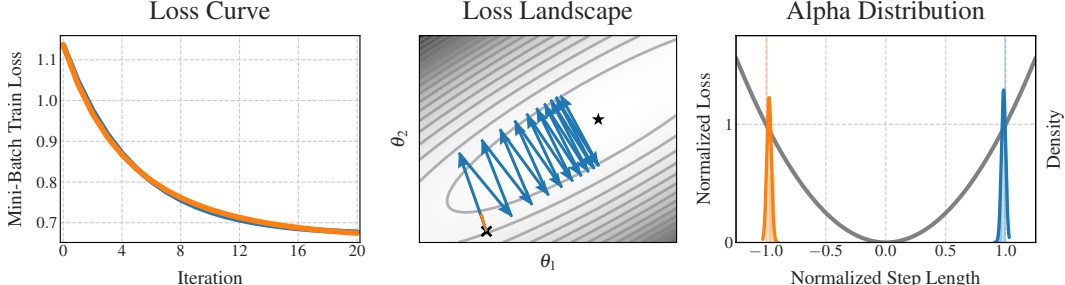

Figure 1: **Illustrative example: Learning curves do not tell the whole story**. Two different optimization runs (—/—) can lead to virtually the same loss curve (*left*). However, the actual optimization trajectories (*middle*), exhibit vastly different behaviors. In practice, the trajectories are intractably large and cannot be visualized directly. Recommendable actions for both scenarios (increase/decrease the learning rate) cannot be inferred from the loss curve. The $\alpha$-distribution, one COCKPIT instrument (*right*), not only clearly distinguishes the two scenarios, but also allows for taking decisions regarding how the learning rate should be adapted. See Section 3.3 for further details.

towards successful training. Extracting meaningful information requires a statistical approach and distillation of the bewildering complexity into a manageable summary. Tools like TENSORBOARD [1] or WEIGHTS & BIASES [6] were built in part to streamline this visualization. Yet, the quantities that are widely monitored (mainly train/test loss & accuracy), provide only scant explanation for relative differences between multiple training runs, because *they do not show the network's internal state*. Figure 1 illustrates how such established learning curves can describe the *current* state of the model – whether it is performing well or not – while failing to inform about training state and dynamics. They tell the user *that* things are going well or badly, but not *why*. The situation is similar to flying a plane by sight, without instruments to provide feedback. It is not surprising, then, that achieving state-of-the-art performance in deep learning requires expert intuition, or plain trial & error.

We aim to enrich the deep learning pipeline with a visual and statistical debugging tool that uses newly proposed observables as well as several established ones (Section 2). We leverage and augment recent extensions to automatic differentiation (i.e. BACKPACK [12] for PYTORCH [33]) to efficiently access second-order statistical (e.g. gradient variances) and geometric (e.g. Hessian) information. We show how these quantities can aid the deep learning engineer in tasks, like learning rate selection, as well as detecting common bugs with data processing or model architectures (Section 3).

Concretely, we introduce COCKPIT, a flexible and efficient framework for online-monitoring these observables during training in carefully designed plots we call "instruments" (see Figure 2). To be of practical use, such visualization must have a manageable computational overhead. We show that COCKPIT scales well to real-world deep learning problems (see Figure 2 and Section 4). We also provide three different configurations of varying computational complexity and demonstrate that their instruments keep the computational cost *well below* a factor of 2 in run time (Section 5). It is available as open-source code,[2] extendable, and seamlessly integrates into existing PYTORCH training loops (see Appendix A).

## 2 COCKPIT's instruments

**Setting:** We consider supervised regression/classification with labeled data $(\boldsymbol{x}, \boldsymbol{y}) \in \mathbb{X} \times \mathbb{Y}$ generated by a distribution $P(\boldsymbol{x}, \boldsymbol{y})$. The training set $\mathcal{D} = \{(\boldsymbol{x}_n, \boldsymbol{y}_n) \mid n = 1, \ldots, N\}$ consists of $N$ i.i.d. samples from $P$ and the deep model $f : \Theta \times \mathbb{X} \to \mathbb{Y}$ maps inputs $\boldsymbol{x}_n$ to predictions $\hat{\boldsymbol{y}}_n$ by parameters $\boldsymbol{\theta} \in \mathbb{R}^D$. This prediction is evaluated by a loss function $\ell : \mathbb{Y} \times \mathbb{Y} \to \mathbb{R}$ which compares to the label $\boldsymbol{y}_n$. The goal is minimizing an inaccessible expected risk $\mathcal{L}_P(\boldsymbol{\theta}) = \int \ell(f(\boldsymbol{\theta}, \boldsymbol{x}), \boldsymbol{y}) \, \mathrm{d}P(\boldsymbol{x}, \boldsymbol{y})$ by empirical approximation through $\mathcal{L}_{\mathcal{D}}(\boldsymbol{\theta}) = \frac{1}{N} \sum_{n=1}^{N} \ell(f(\boldsymbol{\theta}, \boldsymbol{x}_n), \boldsymbol{y}_n) := \frac{1}{N} \sum_{n=1}^{N} \ell_n(\boldsymbol{\theta})$, which

---

[2]https://github.com/f-dangel/cockpit

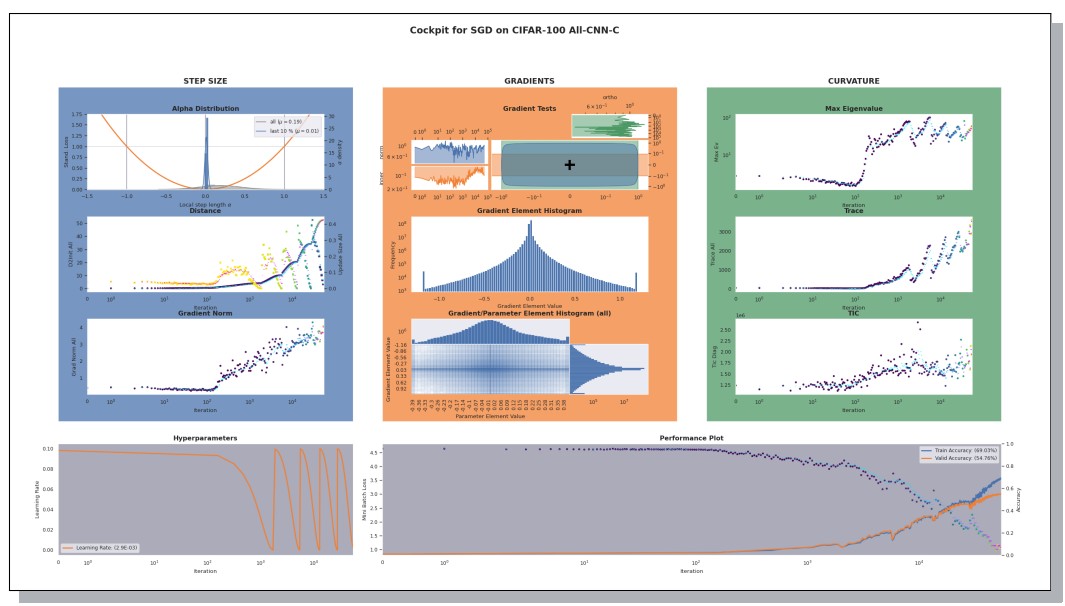

Figure 2: **Screenshot of COCKPIT's full view** while training the ALL-CNN-C [41] on CIFAR-100 with SGD using a cyclical learning rate schedule. This figure and its labels are not meant to be legible, but rather give an impression of COCKPIT's user experience. Gray panels (bottom row) show the information currently tracked by most practitioners. The individual instruments are discussed in Section 2, and observations are described in Section 4. An animated version can be found in the accompanying GitHub repository.

in practice though can only be stochastically sub-sampled on mini-batches $\mathcal{B} \subseteq \{1, \ldots, N\}$,

$$\mathcal{L}_{\mathcal{B}}(\boldsymbol{\theta}) = \frac{1}{|\mathcal{B}|} \sum_{n \in \mathcal{B}} \ell_n(\boldsymbol{\theta}). \tag{1}$$

As is standard practice, we use first- and second-order information of the mini-batch loss, described by its gradient $\boldsymbol{g}_{\mathcal{B}}(\boldsymbol{\theta})$ and Hessian $\boldsymbol{H}_{\mathcal{B}}(\boldsymbol{\theta})$,

$$\boldsymbol{g}_{\mathcal{B}}(\boldsymbol{\theta}) = \frac{1}{|\mathcal{B}|} \sum_{n \in \mathcal{B}} \underbrace{\nabla_{\boldsymbol{\theta}} \ell_n(\boldsymbol{\theta})}_{\boldsymbol{g}_n(\boldsymbol{\theta})}, \qquad \boldsymbol{H}_{\mathcal{B}}(\boldsymbol{\theta}) = \frac{1}{|\mathcal{B}|} \sum_{n \in \mathcal{B}} \nabla_{\boldsymbol{\theta}}^2 \ell_n(\boldsymbol{\theta}). \tag{2}$$

**Design choices:** To minimize computational and design overhead, we restrict the metrics to quantities that require no additional model evaluations. This means that, at training step $t \to t+1$ with mini-batches $\mathcal{B}_t, \mathcal{B}_{t+1}$ and parameters $\boldsymbol{\theta}_t, \boldsymbol{\theta}_{t+1}$, we may access information about the mini-batch losses $\mathcal{L}_{\mathcal{B}_t}(\boldsymbol{\theta}_t)$ and $\mathcal{L}_{\mathcal{B}_{t+1}}(\boldsymbol{\theta}_{t+1})$, but no cross-terms that would require additional forward passes.

**Key point:** $\mathcal{L}_{\mathcal{B}}(\boldsymbol{\theta}), \boldsymbol{g}_{\mathcal{B}}(\boldsymbol{\theta})$, and $\boldsymbol{H}_{\mathcal{B}}(\boldsymbol{\theta})$ are just expected values of a *distribution* over the batch. Only recently, this distribution has begun to attract attention [15] as its computation has become more accessible [8; 12]. Contemporary optimizers leverage only the *mean* gradient and neglect higher moments. One core point of our work is making extensive use of these distribution properties, trying to visualize them in various ways. This out-of-the-box support for the carefully selected and efficiently computed quantities distinguishes COCKPIT from tools like TENSORBOARD that offer visualizations as well. Leveraging these distributional quantities, we create instruments and show how they can help adapt hyperparameters (Section 2.1), analyze the loss landscape (Section 2.2), and track network dynamics (Section 2.3). Instruments can sometimes be built from already-computed information or are efficient variants of previously proposed observables. To keep the presentation concise, we highlight the instruments shown in Figure 2 and listed in Table 1. Appendix C defines them formally and contains more extensions, such as the mean GSNR [27], the early stopping [29] and CABS [4] criterion, which can all be used in COCKPIT.

Table 1: **Overview of COCKPIT quantities**. They range from cheap byproducts, to nonlinear transformations of first-order information and Hessian-based measures. Some quantities have already been proposed, others are first to be considered in this work. They are categorized into configurations *economy* $\subseteq$ *business* $\subseteq$ *full* based on their run time overhead (see Section 5 for a detailed evaluation).

| Name | Short Description | Config | Pos. in Figure 2 |
|---|---|---|---|
| `Alpha` | Normalized step size on a noisy quadratic interpolation between two iterates $\boldsymbol{\theta}_t, \boldsymbol{\theta}_{t+1}$ | *economy* | top left |
| `Distance` | Distance from initialization $\|\boldsymbol{\theta}_t - \boldsymbol{\theta}_0\|_2$ | *economy* | middle left |
| `UpdateSize` | Update size $\|\boldsymbol{\theta}_{t+1} - \boldsymbol{\theta}_t\|_2$ | *economy* | middle left |
| `GradNorm` | Mini-batch gradient norm $\|\boldsymbol{g}_{\mathcal{B}}(\boldsymbol{\theta})\|_2$ | *economy* | bottom left |
| `NormTest` | Normalized fluctuations of the residual norms $\|\boldsymbol{g}_{\mathcal{B}} - \boldsymbol{g}_n\|_2$, proposed in [9] | *economy* | top center |
| `InnerTest` | Normalized fluctuations of the $\boldsymbol{g}_n$'s parallel components along $\boldsymbol{g}_{\mathcal{B}}$, proposed in [7] | *economy* | top center |
| `OrthoTest` | Same as `InnerTest` but using the orthogonal components, proposed in [7] | *economy* | top center |
| `GradHist1d` | Histogram of individual gradient elements, $\{\boldsymbol{g}_n(\boldsymbol{\theta}_j)\}_{n\in\mathcal{B}}^{j=1,...,D}$ | *economy* | middle center |
| `TICDiag` | Relation between (diagonal) curvature and gradient noise, inspired by [43] | *business* | bottom right |
| `HessTrace` | Exact or approximate Hessian trace, $\text{Tr}(\boldsymbol{H}_{\mathcal{B}}(\boldsymbol{\theta}))$, inspired by [50] | *business* | middle right |
| `HessMaxEV` | Maximum Hessian eigenvalue, $\lambda_{\max}(\boldsymbol{H}_{\mathcal{B}}(\boldsymbol{\theta}))$, inspired by [50] | *full* | top right |
| `GradHist2d` | Histogram of weights and individual gradient elements, $\{(\boldsymbol{\theta}_j, \boldsymbol{g}_n(\boldsymbol{\theta}_j))\}_{n\in\mathcal{B}}^{j=1,...,D}$ | *full* | bottom center |

**Bug types:** We distinguish three types of bugs encountered in deep learning. *Implementation bugs* are low-level software bugs that, for example, trigger syntax errors. *Training bugs* result in unnecessarily inefficient or even unsuccessful training. They can, for example, stem from erroneous data handling (see Section 3.1), the chosen model architecture (see Section 3.2), or ill-chosen hyperparameters (see Section 3.3). *Prediction bugs* describe incorrect predictions of a trained model on specific examples. Traditional debuggers are well-suited to find implementation bugs. COCKPIT focuses on efficiently identifying training bugs instead.

## 2.1 Adapting hyperparameters

One big challenge in deep learning is setting the hyperparameters correctly, which is currently mostly done by trial & error through parameter searches. We aim to augment this process with instruments that inform the user about the effect that the chosen parameters have on the current training process.

**Alpha: Are we crossing the valley?** Using individual loss and gradient observations at the start and end point of each iteration, we build a noise-informed univariate quadratic approximation along the step direction (i.e. the loss as a function of the step size), and assess to which point on this parabola our optimizer moves. We standardize this value $\alpha$ such that stepping to the valley-floor is assigned $\alpha = 0$, the starting point is at $\alpha = -1$ and updates to the point exactly opposite of the starting point have $\alpha = 1$ (see Appendix C.2 for a more detailed visual and mathematical description of $\alpha$). Figure 1 illustrates the scenarios $\alpha = \pm 1$ and how monitoring the $\alpha$-distribution (right panel) can help distinguish between two training runs with similar performance but distinct failure sources. By default, this COCKPIT instrument shows the $\alpha$-distribution for the last 10 % of training and the entire training process (e.g. top left plot in Figure 2). In Section 3.3 we demonstrate empirically that, counter-intuitively, it is generally *not* a good idea to choose the step size such that $\alpha$ is close to zero.

**Distances: Are we making progress?** Another way to discern the trajectories in Figure 1 is by measuring the $L_2$ *distance from initialization* [31] and the *update size* [2; 16] in parameter space. Both are shown together in one COCKPIT instrument (see also middle left plot in Figure 2) and are far larger for the blue line in Figure 1. These distance metrics are also able to disentangle phases for the blue path. Using the same step size, it will continue to "jump back and forth" between the loss valley's walls but at some point cease to make progress. During this "surfing of the walls", the *distance from initialization* increases, ultimately though, it will stagnate, with the *update size* remaining non-zero, indicating diffusion. While the initial "surfing the wall"-phase benefits training (see Section 3.3), achieving stationarity may require adaptation once the optimizer reaches that diffusion.

**Gradient norm: How steep is the wall?** The *update size* will show that the orange trajectory is stuck. But why? Such slow-down can result from both a bad learning rate and from loss landscape plateaus. The *gradient norm* (bottom left panel in Figure 2) distinguishes these two causes.

**Gradient tests: How noisy is the batch?** The batch size trades off gradient accuracy versus computational cost. Recently, adaptive sampling strategies based on testing geometric constraints between mean and individual gradients have been proposed [9; 7]. The *norm*, *inner product*, and *orthogonality tests* use a standardized radius and two band widths (parallel and orthogonal to the gradient mean) that indicate how strongly individual gradients scatter around the mean. The original works use these values to adapt batch sizes. Instead, COCKPIT combines all three tests into a single gauge (top center plot of Figure 2) using the standardized noise radius and band widths for visualization. These noise signals can be used to guide batch size adaptation on- and offline, or to probe the influence of gradient alignment on training speed [37] and generalization [10; 11; 27].

## 2.2 Hessian properties for local loss geometry

An intuition for the local loss landscape helps in many ways. It can help diagnose whether training is stuck, to adapt the step size, and explain stability or regularization [18; 23]. The key challenge is the large number of weights: Low-dimensional projections of surfaces can behave unintuitively [30], but tracking the extreme or average behaviors may help in debugging, especially if first-order metrics fail.

**Hessian eigenvalues: A gorge or a lake?** In convex optimization, the maximum Hessian eigenvalue crucially determines the appropriate step size [38]. Many works have studied the Hessian spectrum in machine learning [e.g. 17; 18; 30; 35; 36; 50]. In short: curvature matters. Established [34] and recent autodiff frameworks [12] can compute Hessian properties without requiring the full matrix. COCKPIT leverages this to provide the *Hessian's largest eigenvalue* and *trace* (top right and middle right plots in Figure 2). The former resembles the loss surface's sharpest valley and can thus hint at training instabilities [23]. The *trace* describes a notion of "average curvature", since the eigenvalues $\lambda_i$ relate to it by $\sum_i \lambda_i = \mathrm{Tr}(\boldsymbol{H}_{\mathcal{B}}(\boldsymbol{\theta}))$, which might correlate with generalization [22].

**TIC: How do curvature and gradient noise interact?** There is an ongoing debate about curvature's link to generalization [e.g. 14; 21; 24]. The *Takeuchi Information Criterion (TIC)* [42; 43] estimates the generalization gap by a ratio between Hessian and non-central second gradient moment. It also provides intuition for changes in the objective function implied by gradient noise. Inspired by the approximations in [43], COCKPIT provides mini-batch TIC estimates (bottom right plot of Figure 2).

## 2.3 Visualizing internal network dynamics

Histograms are a natural visual compression of the high-dimensional $|\mathcal{B}| \times D$ individual gradient values. They give insights into the gradient *distribution* and hence offer a more detailed view of the learning signal. Together with the parameter associated to each individual gradient, the entire model status and dynamics can be visualized in a single plot and be monitored during training. This provides a more fine-grained view of training compared to tracking parameters and gradient norms [16].

**Gradient and parameter histograms: What is happening in our network?** COCKPIT offers a univariate *histogram of the gradient elements* $\{\boldsymbol{g}_n(\boldsymbol{\theta})_j\}_{n\in\mathcal{B}}^{j=1,\ldots,D}$. Additionally, a combined *histogram of parameter-gradient pairs* $\{(\boldsymbol{\theta}_j, \boldsymbol{g}_n(\boldsymbol{\theta})_j)\}_{n\in\mathcal{B}}^{j=1,\ldots,D}$ provides a two-dimensional look into the network's gradient and parameter values in a mini-batch. Section 3.1 shows an example use-case of the gradient histogram; Section 3.2 makes the case for the layer-wise variants of the instruments.

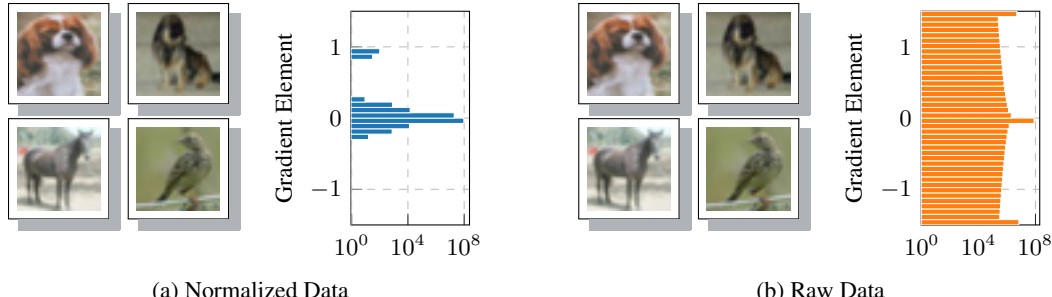

(a) Normalized Data                                    (b) Raw Data

Figure 3: **Same inputs, different gradients; Catching data bugs with COCKPIT.** (a) *normalized* ($[0, 1]$) and (b) *raw* ($[0, 255]$) images look identical in auto-scaled front-ends like MATPLOTLIB's `imshow`. The gradient distribution on the 3C3D model, however, is crucially affected by this scaling.

## 3   Experiments

The diverse information provided by COCKPIT can help users and researchers in many ways, some of which, just like for a traditional debugger, only become apparent in practical use. In this section, we present a few motivating examples, selecting specific instruments and scenarios in which they are practically useful. Specifically, we show that COCKPIT can help the user discern between, and thus fix, common training bugs (Sections 3.1 and 3.2) that are otherwise hard to distinguish as they lead to the same failure: bad training. We demonstrate that COCKPIT can guide practitioners to choose efficient hyperparameters *within a single training run* (Sections 3.2 and 3.3). Finally, we highlight that COCKPIT's instruments can provide research insights about the optimization process (Section 3.3). Our empirical findings are demonstrated on problems from the DEEPOBS [39] benchmark collection.

### 3.1   Incorrectly scaled data

One prominent source of bugs is the data pipeline. To pick a relatively simple example: For standard optimizers to work at their usual learning rates, network inputs must be standardized (i.e. between zero and one, or have zero mean and unit variance [e.g. 5]). If the user forgets to do this, optimizer performance is likely to degrade. It can be difficult to identify the source of this problem as it does not cause obvious failures, `NaN` or `Inf` gradients, etc. We now construct a semi-realistic example, to show how using COCKPIT can help diagnose this problem upon observing slow training performance.

By default[3], the popular image data sets CIFAR-10/100 [25] are provided as NUMPY [20] arrays that consist of integers in the interval $[0, 255]$. This *raw* data, instead of the widely used version with floats in $[0, 1]$, changes the data scale by a factor of 255 and thus the gradients as well. Therefore, the optimizer's optimal learning rate is scaled as well. In other words, the default parameters of popular optimization methods may not work well anymore, or good hyperparameters may take extreme values. Even if the user directly inspects the training images, this may not be apparent (see Figure 3 and Figure 10 in the appendix for the same experiment with VGG16 on IMAGENET). But the gradient histogram instrument of COCKPIT, which has a deliberate default plotting range around $[-1, 1]$ to highlight such problems, immediately and prominently shows that there is an issue.

Of course, this particular data is only a placeholder for real practical data sets. While this problem may not frequently arise in the highly pre-processed, packaged CIFAR-10, it is not a rare problem for practitioners who work with their personal data sets. This is particularly likely in domains outside standard computer vision, e.g. when working with mixed-type data without obvious natural scales.

### 3.2   Vanishing gradients

The model architecture itself can be a source of training bugs. As before, such problems mostly arise with novel data sets, where well-working architectures are unknown. The following example shows how even small (in terms of code) architecture modifications may severely harm the training.

---

[3] https://www.cs.toronto.edu/~kriz/cifar.html

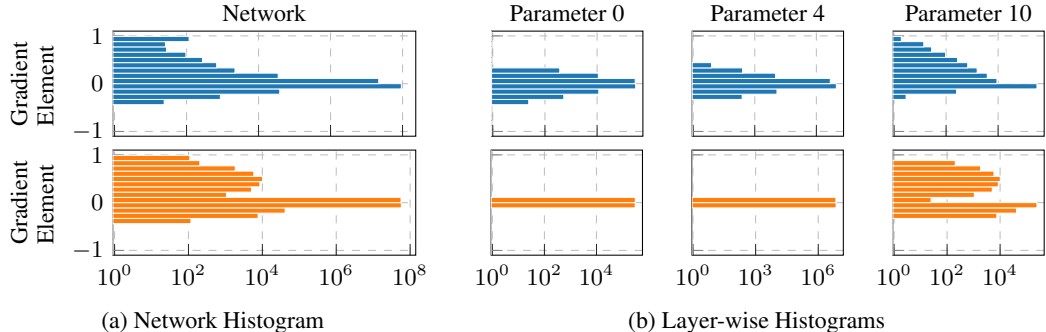

(a) Network Histogram      (b) Layer-wise Histograms

Figure 4: **Gradient distributions of two similar architectures on the same problem**. (a) Distribution of individual gradient elements summarized over the entire network. Both seem similar. (b) Layer-wise histograms for a subset of layers. Parameter 0 is the layer closest to the network's input, parameter 10 closest to its output. Only the layer-wise view reveals that there are several degenerated gradient distributions for the orange network making training unnecessary hard.

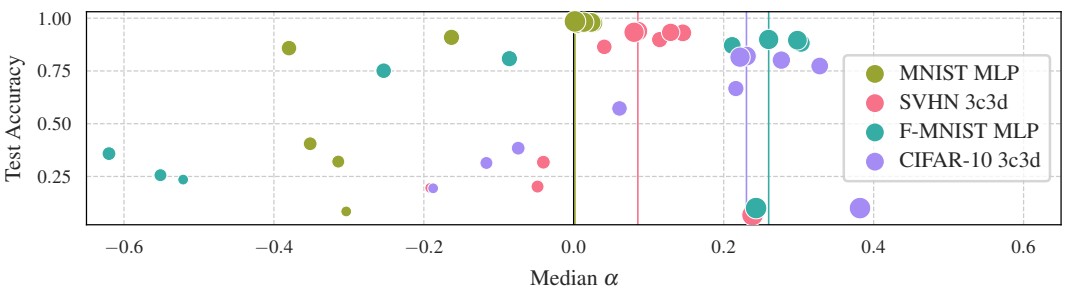

Figure 5: **Test accuracy as a function of standardized step size** $\alpha$. For four DEEPOBS problems (see Appendix E), final test accuracy is shown versus the median $\alpha$-value over the entire training. Marker size indicates the magnitude of the raw learning rate, marker color identifies tasks (see legend). For each problem, the best-performing setting is highlighted by a vertical colored line.

Figure 4a shows the distribution of gradient values of two different network architectures in blue and orange. Although the blue model trains considerably better than the orange one, their gradient distributions look quite similar. The difference becomes evident when inspecting the histogram *layer-wise*. We can see that multiple layers have a degenerated gradient distribution with many elements being practically zero (see Figure 4b, bottom row). Since the fully connected layers close to the output have far more parameters (a typical pattern of convolutional networks), they dominate the network-wide histogram. This obscures that a major part of the model is effectively unable to train.

Both the blue and orange networks follow DEEPOBS's 3C3D architecture. The only difference is the non-linearity: The blue network uses standard ReLU activations, while the orange one has sigmoid activations. Here, the layer-wise histogram instrument of COCKPIT highlights which part of the architecture makes training unnecessarily hard. Accessing information layer-wise is also essential due to the strong overparameterization in deep models where training can happen in small subspaces [19]. Once again, this is hard to do with common monitoring tools, such as the loss curve.

### 3.3 Tuning learning rates

Once the architecture is defined, the optimizer's learning rate is the most important hyperparameter to tune. Getting it right requires extensive hyperparameter searches at high resource costs. COCKPIT's instruments can provide intuition and information to streamline this process: In contrast to the raw learning rate, the curvature-standardized step size $\alpha$-quantity (see Section 2.1) has a natural scale.

Across multiple optimization problems, we observe, perhaps surprisingly, that the best runs and indeed all good runs have a median $\alpha > 0$ (Figure 5). This illustrates a fundamental difference between stochastic optimization, as is typical for machine learning, and classic deterministic optimization. Instead of locally stepping "to the valley floor" (optimal in the deterministic case), stochastic

optimizers should *overshoot* the valley somewhat. This need to "surf the walls" has been hypothesized before [e.g. 47; 49] as a property of neural network training. Frequently, learning rates are adapted during training, which fits with our observation about positive $\alpha$-values: "Overshooting" allows fast early progression towards areas of lower loss, but it does not yield convergence in the end. Real-time visualizations of the training state, as offered by COCKPIT, can augment these fine-tuning processes.

Figure 5 also indicates a major challenge preventing simple automated tuning solutions: The optimal $\alpha$-value is problem-dependent, and simpler problems, such as a multi-layer perceptron (MLP) on MNIST [26], behave much more similar to classic optimization problems. Algorithmic research on small problems can thus produce misleading conclusions. The figure also shows that the $\alpha$-gauge is not sufficient by itself: extreme overshooting with a too-large learning rate leads to poor performance, which however can be prevented by taking additional instruments into account. This makes the case for the cockpit metaphor of increasing interpretability from several instruments in conjunction. By combining the $\alpha$-instrument with other gauges that capture the local geometry or network dynamics, the user can better identify good choices of the learning rate and other hyperparameters.

## 4    Showcase

Having introduced the tool, we can now return to Figure 2 for a closer look. The figure shows a snapshot from training the ALL-CNN-C [41] on CIFAR-100 using SGD with a cyclic learning rate schedule (see bottom left panel). Diagonal curvature instruments are configured to use an MC approximation in order to reduce the run time (here, $C = 100$, compare Section 5).

A glance at all panels shows that the learning rate schedule is reflected in the metrics. However, the instruments also provide insights into the early phase of training (first $\sim 100$ iterations), where the learning rate is still unaffected by the schedule: There, the loss plateaus and the optimizer takes relatively small steps (compared to later, as can be seen in the small gradient norms, and small distance from initialization). Based on these low-cost instruments, one may thus at first suspect that training was poorly initialized; but training indeed succeeds after iteration 100! Viewing COCKPIT entirely though, it becomes clear that optimization in these first steps is not stuck at all: While loss, gradient norms, and distance in parameter space remain almost constant, curvature changes, which expresses itself in a clear downward trend of the maximum Hessian eigenvalue (top right panel).

The importance of early training phases has recently been hypothesized [16], suggesting a logarithmic timeline. Not only does our showcase support this hypothesis, but it also provides an explanation from the curvature-based metrics, which in this particular case are the only meaningful feedback in the first few training steps. It also suggests monitoring training at log-spaced intervals. COCKPIT provides the flexibility to do so, indeed, Figure 2 has been created with log-scheduled tracking events.

As a final note, we recognize that the approach taken here promotes an amount of *manual* work (monitoring metrics, deliberately intervening, etc.) that may seem ironic and at odds with the paradigm of automation that is at the heart of machine learning. However, we argue that this might be what is needed at this point in the evolution of the field. Deep learning has been driven notably by scaling compute resources [44], and fully automated, one-shot training may still be some way out. To develop better training methods, researchers, not just users, need *algorithmic* interpretability and explainability: direct insights and intuition about the processes taking place "inside" neural nets. To highlight how COCKPIT might provide this, we contrast in Appendix F the COCKPIT view of two convex DEEPOBS problems: a noisy quadratic and logistic regression on MNIST. In both cases, the instruments behave differently compared to the deep learning problem in Figure 2. In particular, the gradient norm increases (left column, bottom panel) during training, and individual gradients become less scattered (center column, top panel). This is diametrically opposed to the convex problems and shows that deep learning differs even qualitatively from well-understood optimization problems.

## 5    Benchmark

Section 3 made a case for COCKPIT as an effective debugging and tuning tool. To make the library useful in practice, it must also have limited computational cost. We now show that it is possible to compute all quantities at reasonable overhead. The user can control the absolute cost along two dimensions, by reducing the number of instruments, or by reducing their update frequency.

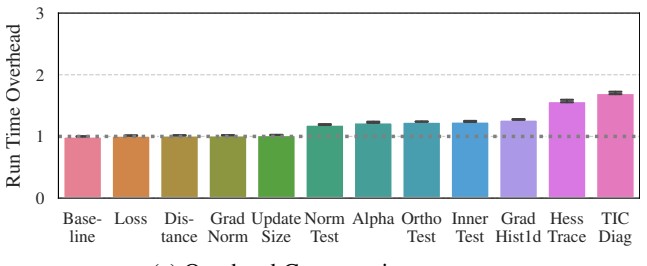

(a) Overhead Cockpit instruments

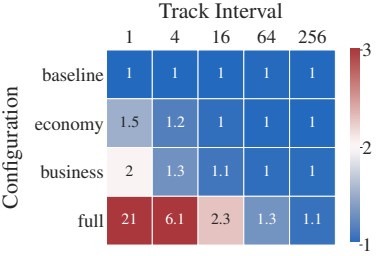

(b) Overhead Cockpit configurations

Figure 6: **Run time overhead for individual Cockpit instruments and configurations** as shown on CIFAR-10 3c3d on a GPU. *Left:* The run time overheads for individual instruments are shown as multiples of the *baseline* (no tracking). Most instruments add little overhead. This plot shows the overhead in one iteration, determined by averaging over multiple iterations and random seeds. *Right:* Overhead for different Cockpit configurations. Adjusting the tracking interval and re-using the computation shared by multiple instruments can make the overhead orders of magnitude smaller. Blue fields mark settings that allow tracking without doubling the training time.

All benchmark results show SGD without momentum. Cockpit's quantities, however, work for generic optimizers and can mostly be used identically without increased costs. One current exception is `Alpha` which can be computed more efficiently given the optimizer's update rule.[4]

**Complexity analysis:** Computing more information adds computational overhead, of course. However, recent work [12] has shown that first-order information, like distributional statistics on the batch gradients, can be computed on top of the mean gradient at little extra cost. Similar savings apply for most quantities in Table 1, as they are (non-)linear transformations of individual gradients. A subset of Cockpit's quantities also uses second-order information from the Hessian diagonal. For ReLU networks on a classification task with $C$ classes, the additional work is proportional to $C$ gradient backpropagations (i.e. $C = 10$ for CIFAR-10, $C = 100$ for CIFAR-100). Parallel processing can, to some extent, process these extra backpropagations in parallel without significant overhead. If this is no longer possible, we can fall back to a Monte Carlo (MC) sampling approximation, which reduces the number of extra backprop passes to the number of samples (1 by default).[5]

While parallelization is possible for the gradient instruments, computing the maximum Hessian eigenvalue is inherently sequential. Similar to Yao et al. [50], we use matrix-free Hessian-vector products by automatic differentiation [34], where each product's costs are proportional to one gradient computation. Regardless of the underlying iterative eigensolver, multiple such products must be queried to compute the spectral norm (the required number depends on the spectral gap to the second-largest eigenvalue).

**Run time benchmark:** Figure 6a shows the wall-clock computational overhead for individual instruments (details in Appendix E).[6] As expected, byproducts are virtually free, and quantities that rely solely on first-order information add little overhead (at most roughly 25 % on this problem). Thanks to parallelization, the ten extra backward passes required for Hessian quantities reduce to less than 100 % overhead. Individual overheads also do not simply add up when multiple quantities are tracked, because quantities relying on the same information share computations.

To allow a rough cost control, Cockpit currently offers three configurations, called *economy*, *business*, and *full*, in increasing order of cost (cf. Table 1). As a basic guideline, we consider a factor of two to be an acceptable limit for the increase in training time and benchmark the configurations'

---

[4]This is currently implemented for vanilla SGD. Otherwise, Cockpit falls back to a less efficient scheme.

[5]An MC-sampled approximation of the Hessian/generalized Gauss-Newton has been used in Figure 2 to reduce the prohibitively large number of extra backprops on CIFAR-100 ($C = 100$).

[6]To improve readability, we exclude `HessMaxEV` here, because its overhead is large compared to other quantities. Surprisingly, we also observed significant cost for the 2D histogram on GPU. It is caused by an implementation bottleneck for histogram shapes observed in deep models. We thus also omit `GradHist2d` here, as we expect it to be eliminated with future implementations (see Appendix E.2 for a detailed analysis and further benchmarks). Both quantities, however, are part of the benchmark shown in Figure 6b.

run times for different tracking intervals. Figure 6b shows a run time matrix for the CIFAR-10 3C3D problem, where settings that meet this limit are set in blue (more problems including IMAGENET are shown in Appendix E). Speedups due to shared computations are easy to read off: Summing all the individual overheads shown in Figure 6a would result in a total overhead larger than 200 %, while the joint overhead (*business*) reduces to 140 %. The *economy* configuration can easily be tracked at every step of this problem and stay well below our threshold of doubling the execution time. COCKPIT's full view, shown in Figure 2, can be updated every 64-th iteration without a major increase in training time (this corresponds to about five updates per epoch). Finally, tracking any configuration about once per epoch – which is common in practice – adds overhead close to zero (rightmost column).

This good performance is largely due to the efficiency of the BACKPACK package [12], which we leverage with custom and optimized modification, that compacts information layer-wise and then discards unneeded buffers. Using layer-wise information (Section 3.2) scales better to large networks, where storing the entire model's individual gradients all at once becomes increasingly expensive (see Appendix E). To the best of our knowledge, many of the quantities in Table 1, especially those relying on individual gradients, have only been explored on rather small problems. With COCKPIT they can now be accessed at a reasonable rate for deep learning models outside the toy problem category.

## 6 Conclusion

Contemporary machine learning, in particular deep learning, remains a craft and an art. High dimensionality, stochasticity, and non-convexity require constant tracking and tuning, often resulting in a painful process of trial and error. When things fail, popular performance measures, like the training loss, do not provide enough information by themselves. These metrics only tell *whether* the model is learning, but not *why*. Alternatively, traditional debugging tools can provide access to individual weights and data. However, in models whose power only arises from possessing myriad weights, this approach is hopeless, like looking for the proverbial needle in a haystack.

To mitigate this, we proposed COCKPIT, a practical visual debugging tool for deep learning. It offers instruments to monitor the network's internal dynamics during training, in real-time. In its presentation, we focused on two crucial factors affecting user experience: Firstly, such a debugger must provide meaningful insights. To demonstrate COCKPIT's utility, we showed how it can identify bugs where traditional tools fail. Secondly, it must come at a feasible computational cost. Although COCKPIT uses rich second-order information, efficient computation keeps the necessary run time overhead cheap. The open-source PYTORCH package can be added to many existing training loops.

Obviously, such a tool is never complete. Just like there is no perfect universal debugger, the list of current instruments is naturally incomplete. Further practical experience with the tool, for example in the form of a future larger user study, could provide additional evidence for its utility. However, our analysis shows that COCKPIT provides useful tools and extracts valuable information presently not accessible to the user. We believe that this improves algorithmic interpretability – helping practitioners understand how to make their models work – but may also inspire new research. The code is designed flexibly, deliberately separating the computation and visualization. New instruments can be added easily and also be shown by the user's preferred visualization tool, e.g. TENSORBOARD. Of course, instead of just showing the data, the same information can be used by novel algorithms directly, side-stepping the human in the loop.

### Acknowledgments and Disclosure of Funding

The authors gratefully acknowledge financial support by the European Research Council through ERC StG Action 757275 / PANAMA; the DFG Cluster of Excellence "Machine Learning - New Perspectives for Science", EXC 2064/1, project number 390727645; the German Federal Ministry of Education and Research (BMBF) through the Tübingen AI Center (FKZ: 01IS18039A); and funds from the Cyber Valley Initiative of the Ministry for Science, Research and Arts of the State of Baden-Württemberg. Moreover, the authors thank the International Max Planck Research School for Intelligent Systems (IMPRS-IS) for supporting Felix Dangel and Frank Schneider. Further, we are grateful to Agustinus Kristiadi, Alexandra Gessner, Christian Fröhlich, Filip de Roos, Jonathan Wenger, Julia Grosse, Lukas Tatzel, Marius Hobbhahn, and Nicholas Krämer for providing feedback to the manuscript.

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
