# COCKPIT: A Practical Debugging Tool
# for the Training of Deep Neural Networks
## Supplementary Material


# A    Code example

One design principle of COCKPIT is its easy integration with conventional PYTORCH training loops. Figure 7 shows a working example of a standard training loop with COCKPIT integration. More examples and tutorials are described in COCKPIT's documentation. COCKPIT's syntax is inspired by BACKPACK: It can be used interchangeably with the library responsible for most back-end computations. Changes to the code are straightforward:

- **Importing** (*Lines 5, 7* and *8*): Besides importing COCKPIT we also need to import BACKPACK which is required for extending (parts of) the model (see next step).

- **Extending** (*Lines 11* and *12*): When defining the model and the loss function, we need to *extend* both of them using BACKPACK. This is as trivial as wrapping them in the `extend()` function provided by BACKPACK and lets BACKPACK know that additional quantities (such as the individual gradients) should be computed for them. Note, that while applying BACKPACK is easy, it currently does not support all possible model architectures and layer types. Specifically, *batch norm* layers are not supported since using them results in ill-defined individual gradients.

- **Individual losses** (*Line 13*): For the `Alpha` quantity, COCKPIT also requires the individual loss values (to estimate the variance of the loss estimate). This can be computed cheaply but is not usually part of a conventional training loop. Creating this loss is done analogously to creating any other loss, with the only exception of setting `reduction="none"`. Since we don't differentiate this loss, we don't need to extend it.

- **Cockpit configuration** (*Line 16* and *17*): Initializing the COCKPIT requires passing them (extended) model parameters as well as a list of quantities that should be tracked. Table 1 provides an overview of all possible quantities. In this example, we use one of the predefined configurations offered by COCKPIT. Separately, we initialize the plotting part of COCKPIT. We deliberately detached the visualization from the tracking to allow greater flexibility.

- **Quantity computation** (*Line 27* and *38*): Performing the training is very similar to a regular training loop, with the only difference being that the backward pass should be surrounded by the COCKPIT context (`with cockpit():`). Additionally to the `global_step` we also pass a few additional information to the COCKPIT that are computed anyway and can be re-used by the COCKPIT, such as the batch size, the individual losses, or the optimizer itself. After the backward pass (when the context is left) all COCKPIT quantities are automatically computed.

- **Logging and visualizing** (*Line 46* and *47*): At any point during the training, here we do it at the end, we can write all quantities to a log file. We can use this log file, or alternatively the COCKPIT directly, to visualize all quantities which would result in a status screen similar to Figure 2.

```python
1  """Example: Training Loop using Cockpit."""
2
3  import torch
4  from _utils_examples import cnn, fmnist_data, get_logpath
5  from backpack import extend
6  from cockpit import Cockpit, CockpitPlotter
7  from cockpit.utils.configuration import configuration as config
8
9  fmnist_data = fmnist_data()
10 model = extend(cnn())
11 loss_fn = extend(torch.nn.CrossEntropyLoss(reduction="mean"))
12 losses_fn = torch.nn.CrossEntropyLoss(reduction="none")
13 opt = torch.optim.SGD(model.parameters(), lr=1e-2)
14
15 cockpit = Cockpit(model.parameters(), quantities=config("full"))
16 plotter = CockpitPlotter()
17
18 max_steps, global_step = 50, 0
19 for inputs, labels in iter(fmnist_data):
20     opt.zero_grad()
21
22     outputs = model(inputs)
23     loss = loss_fn(outputs, labels)
24     losses = losses_fn(outputs, labels)
25
26     with cockpit(
27         global_step,
28         info={
29             "batch_size": inputs.shape[0],
30             "individual_losses": losses,
31             "loss": loss,
32             "optimizer": opt,
33         },
34     ):
35         loss.backward(
36             create_graph=cockpit.create_graph(global_step),
37         )
38
39     opt.step()
40     global_step += 1
41
42     if global_step >= max_steps:
43         break
44
45 cockpit.write(get_logpath())
46 plotter.plot(get_logpath())
```

Figure 7: **Complete training loop with COCKPIT** in PYTORCH. Line changes are highlighted in light orange (●).

# B COCKPIT instruments overview

Table 2 lists all quantities available in the first public release of COCKPIT. If necessary, we provide references to their mathematical definition. This table contains additional quantities, compared to Table 1 in the main text. To improve the presentation of this work, we decided to not describe every quantity available in COCKPIT in the main part and instead focus on the investigated metrics. Custom quantities can be added easily without having to understand the inner-workings.

Table 2: **Overview of all COCKPIT quantities** with a short description and, if necessary, a reference to mathematical definition.

| Name | Description | Math |
|------|-------------|------|
| Loss | Mini-batch training loss at current iteration, $\mathcal{L}_\mathcal{B}(\boldsymbol{\theta})$ | (1) |
| Parameters | Parameter values $\boldsymbol{\theta}_t$ at the current iteration | - |
| Distance | $L_2$ distance from initialization $\|\boldsymbol{\theta}_t - \boldsymbol{\theta}_0\|_2$ | - |
| UpdateSize | Update size of the current iteration $\|\boldsymbol{\theta}_{t+1} - \boldsymbol{\theta}_t\|_2$ | |
| GradNorm | Mini-batch gradient norm $\|\boldsymbol{g}_\mathcal{B}(\boldsymbol{\theta})\|_2$ | - |
| Time | Time of the current iteration (e.g. used in benchmark of Appendix E) | - |
| Alpha | Normalized step on a noisy quadratic interpolation between two iterates $\boldsymbol{\theta}_t, \boldsymbol{\theta}_{t+1}$ | (9) |
| CABS | Adaptive batch size for SGD, optimizes expected objective gain per cost, adapted from [4] | (11) |
| EarlyStopping | Evidence-based early stopping criterion for SGD, proposed in [29] | (13d) |
| GradHist1d | Histogram of individual gradient elements, $\{\boldsymbol{g}_n(\boldsymbol{\theta}_j)\}_{n\in\mathcal{B}}^{j=1,\ldots,D}$ | (14) |
| GradHist2d | Histogram of weights and individual gradient elements, $\{(\boldsymbol{\theta}_j, \boldsymbol{g}_n(\boldsymbol{\theta}_j))\}_{n\in\mathcal{B}}^{j=1,\ldots,D}$ | (15) |
| NormTest | Normalized fluctuations of the residual norms $\|\boldsymbol{g}_\mathcal{B} - \boldsymbol{g}_n\|$, proposed in [9] | (18c) |
| InnerTest | Normalized fluctuations of $\boldsymbol{g}_n$'s parallel components along $\boldsymbol{g}_\mathcal{B}$, proposed in [7] | (21c) |
| OrthoTest | Normalized fluctuations of $\boldsymbol{g}_n$'s orthogonal components along $\boldsymbol{g}_\mathcal{B}$, proposed in [7] | (24b) |
| HessMaxEV | Maximum Hessian eigenvalue, $\lambda_{\max}(\boldsymbol{H}_\mathcal{B}(\boldsymbol{\theta}))$, inspired by [50] | (25) |
| HessTrace | Exact or approximate Hessian trace, $\mathrm{Tr}(\boldsymbol{H}_\mathcal{B}(\boldsymbol{\theta}))$, inspired by [50] | - |
| TICDiag | Relation between (diagonal) curvature and gradient noise, inspired by [43] | (28) |
| TICTrace | Relation between curvature and gradient noise trace, inspired by [43] | (27) |
| MeanGSNR | Average gradient signal-to-noise-ratio (GSNR), inspired by [27] | (30b) |

## C Mathematical details

In this section, we want to provide the mathematical background for each instrument described in Table 2. This complements the more informal description presented in Section 2 in the main text, which focused more on the expressiveness of the individual quantities. We will start by setting up the necessary notation in addition to the one introduced in Section 2.

### C.1 Additional notation

**Population properties:** The population risk $\mathcal{L}_P(\boldsymbol{\theta}) \in \mathbb{R}$ and its variance $\Lambda(\boldsymbol{\theta}) \in \mathbb{R}$ are given by

$$\mathcal{L}_P(\boldsymbol{\theta}) = \mathbb{E}_{(\boldsymbol{x},\boldsymbol{y})\sim P}\left[\ell(f(\boldsymbol{\theta},\boldsymbol{x}),\boldsymbol{y})\right] = \int \ell(f(\boldsymbol{\theta},\boldsymbol{x}),\boldsymbol{y})P(\boldsymbol{x},\boldsymbol{y})\,d\boldsymbol{x}\,d\boldsymbol{y}\,, \tag{3a}$$

$$\Lambda_P(\boldsymbol{\theta}) = \mathrm{Var}_{(\boldsymbol{x},\boldsymbol{y})\sim P}\left[\ell(f(\boldsymbol{\theta},\boldsymbol{x}),\boldsymbol{y})\right] = \int \left(\ell(f(\boldsymbol{\theta},\boldsymbol{x}),\boldsymbol{y}) - \mathcal{L}_P(\boldsymbol{\theta})\right)^2 P(\boldsymbol{x},\boldsymbol{y})\,d\boldsymbol{x}\,d\boldsymbol{y}\,. \tag{3b}$$

The population gradient $\boldsymbol{g}_P(\boldsymbol{\theta}) \in \mathbb{R}^D$ and its variance $\boldsymbol{\Sigma}_P(\boldsymbol{\theta}) \in \mathbb{R}^{D\times D}$ are given by

$$\boldsymbol{g}_P(\boldsymbol{\theta}) = \mathbb{E}_{(\boldsymbol{x},\boldsymbol{y})\sim P}\left[\nabla_{\boldsymbol{\theta}}\ell(f(\boldsymbol{\theta},\boldsymbol{x}),\boldsymbol{y})\right] = \int \nabla_{\boldsymbol{\theta}}\ell(f(\boldsymbol{\theta},\boldsymbol{x}),\boldsymbol{y})P(\boldsymbol{x},\boldsymbol{y})\,d\boldsymbol{x}\,d\boldsymbol{y}\,, \tag{4a}$$

$$\begin{aligned}\boldsymbol{\Sigma}_P(\boldsymbol{\theta}) &= \mathrm{Var}_{(\boldsymbol{x},\boldsymbol{y})\sim P}\left[\nabla_{\boldsymbol{\theta}}\ell(f(\boldsymbol{\theta},\boldsymbol{x}),\boldsymbol{y})\right]\\ &= \int \left(\nabla_{\boldsymbol{\theta}}\ell(f(\boldsymbol{\theta},\boldsymbol{x}),\boldsymbol{y}) - \boldsymbol{g}_P(\boldsymbol{\theta})\right)\left(\nabla_{\boldsymbol{\theta}}\ell(f(\boldsymbol{\theta},\boldsymbol{x}),\boldsymbol{y}) - \boldsymbol{g}_P(\boldsymbol{\theta})\right)^\top P(\boldsymbol{x},\boldsymbol{y})\,d\boldsymbol{x}\,d\boldsymbol{y}\,.\end{aligned} \tag{4b}$$

**Empirical approximations:** Let $\mathcal{S}$ denote a set of samples drawn i.i.d. from $P$, i.e. $\mathcal{S} = \{(\boldsymbol{x}_i,\boldsymbol{y}_i)\,|\,i=1,\ldots,|\mathcal{S}|\}$. With a slight abuse of notation the empirical risk approximated with $\mathcal{S}$ is

$$\mathcal{L}_\mathcal{S}(\boldsymbol{\theta}) = \frac{1}{|\mathcal{S}|}\sum_{n\in\mathcal{S}}\ell_n(\boldsymbol{\theta}) \tag{5a}$$

(later, $\mathcal{S}$ will represent either a mini-batch $\mathcal{B}$, or the train set $\mathcal{D}$). The empirical risk gradient $\boldsymbol{g}_\mathcal{S}(\boldsymbol{\theta}) \in \mathbb{R}^D$ on $\mathcal{S}$ is

$$\boldsymbol{g}_\mathcal{S}(\boldsymbol{\theta}) = \nabla_{\boldsymbol{\theta}}\mathcal{L}_\mathcal{S}(\boldsymbol{\theta}) = \frac{1}{|\mathcal{S}|}\sum_{n\in\mathcal{S}}\nabla_{\boldsymbol{\theta}}\ell_n(\boldsymbol{\theta}) = \frac{1}{|\mathcal{S}|}\sum_{n\in\mathcal{S}}\boldsymbol{g}_n(\boldsymbol{\theta})\,, \tag{5b}$$

with individual gradients $\boldsymbol{g}_n(\boldsymbol{\theta}) = \nabla_{\boldsymbol{\theta}}\ell_n(\boldsymbol{\theta}) \in \mathbb{R}^D$ implied by a sample $n$. Population risk and gradient variances $\Lambda_P(\boldsymbol{\theta}), \boldsymbol{\Sigma}_P(\boldsymbol{\theta})$ can be empirically estimated on $\mathcal{S}$ with the sample variances $\hat{\Lambda}_\mathcal{S}(\boldsymbol{\theta}) \in \mathbb{R}, \hat{\boldsymbol{\Sigma}}_\mathcal{S}(\boldsymbol{\theta}) \in \mathbb{R}^{D\times D}$, given by

$$\Lambda_P(\boldsymbol{\theta}) \approx \frac{1}{|S|-1}\sum_{n\in\mathcal{S}}\left(\ell_n(\boldsymbol{\theta}) - \mathcal{L}_\mathcal{S}(\boldsymbol{\theta})\right)^2 := \hat{\Lambda}_\mathcal{S}(\boldsymbol{\theta})\,, \tag{6a}$$

$$\begin{aligned}\boldsymbol{\Sigma}_P(\boldsymbol{\theta}) &\approx \frac{1}{|S|-1}\sum_{n\in\mathcal{S}}\left(\boldsymbol{g}_n(\boldsymbol{\theta}) - \boldsymbol{g}_\mathcal{S}(\boldsymbol{\theta})\right)\left(\boldsymbol{g}_n(\boldsymbol{\theta}) - \boldsymbol{g}_\mathcal{S}(\boldsymbol{\theta})\right)^\top := \hat{\boldsymbol{\Sigma}}_\mathcal{S}(\boldsymbol{\theta})\\ &\approx \frac{1}{|S|-1}\left[\left(\sum_{n\in\mathcal{S}}\boldsymbol{g}_n(\boldsymbol{\theta})\boldsymbol{g}_n(\boldsymbol{\theta})^\top\right) - |\mathcal{S}|\boldsymbol{g}_\mathcal{S}(\boldsymbol{\theta})\boldsymbol{g}_\mathcal{S}(\boldsymbol{\theta})^\top\right]\,.\end{aligned} \tag{6b}$$

Often, gradient elements are assumed independent and hence their variance is diagonal ($\odot2$ denotes element-wise square),

$$\mathrm{diag}(\boldsymbol{\Sigma}_P(\boldsymbol{\theta})) \approx \frac{1}{|S|-1}\sum_{n\in\mathcal{S}}\left(\boldsymbol{g}_n(\boldsymbol{\theta}) - \boldsymbol{g}_\mathcal{S}(\boldsymbol{\theta})\right)^{\odot2} = \mathrm{diag}\left(\hat{\boldsymbol{\Sigma}}_\mathcal{S}(\boldsymbol{\theta})\right) \in \mathbb{R}^D\,. \tag{7}$$

**Slicing:** To avoid confusion between $\boldsymbol{\theta}_t$ (parameter at iteration $t$) and $\boldsymbol{\theta}_j$ ($j$-th parameter entry), we denote the latter as $[\boldsymbol{\theta}]_j$.

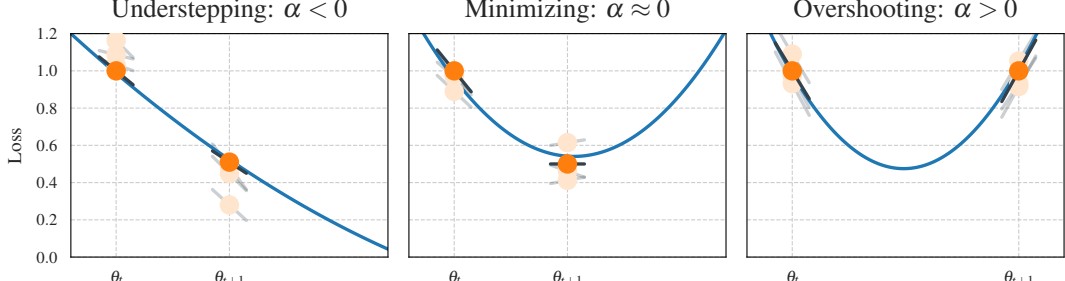

Figure 8: **Motivational sketch for the $\alpha$ quantity.** In each iteration of the optimizer we observe the loss function at two positions $\boldsymbol{\theta}_t$ and $\boldsymbol{\theta}_{t+1}$ (shown in ●). The black lines (—) show the observed slope at this position, which we can get from projecting the gradients onto the current step direction $\boldsymbol{\theta}_{t+1} - \boldsymbol{\theta}_t$. Note, that all four observations (two loss and two slope values) are noisy, due to being computed on a mini-batch. With access to the individual losses and gradients (some samples shown in ●/—), we can estimate their noise level and build a noise-informed quadratic fit (—). Using this fit, we determine whether the optimizer minimizes the local univariate loss (*middle plot*), or whether we understep (*left plot*) or overshoot (*right plot*) the minimum.

## C.2   Normalized Step Length (`Alpha`)

**Motivation:**   The goal of the $\alpha$-quantity is to estimate and quantify the effect that a selected learning rate has on the optimizer's steps. Let's consider the step that the optimizer takes at training iteration $t$. This parameter update from $\boldsymbol{\theta}_t$ to $\boldsymbol{\theta}_{t+1}$ happens in a one-dimensional space, defined by the update direction $\boldsymbol{\theta}_{t+1} - \boldsymbol{\theta}_t = \boldsymbol{s}_t$. The update direction depends on the update rule of the optimizer, e.g. for SGD with learning rate $\eta$ it is simply $\boldsymbol{s}_t = -\eta \boldsymbol{g}_{\mathcal{B}_t}(\boldsymbol{\theta}_t)$.

We build a noise-informed univariate quadratic approximation along this update step ($\boldsymbol{\theta}_t \to \boldsymbol{\theta}_{t+1}$) based on the two noisy loss function observations at $\boldsymbol{\theta}_t$ and $\boldsymbol{\theta}_{t+1}$ and the two noisy slope observation at these two points. Examining this quadratic fit, we are able to determine where on this parabola our optimizer steps. Standardizing this, we express a step to the minimum of the loss in the update direction as $\alpha = 0$. Analogously, steps that end short of this minimum result in $\alpha < 0$, and a step over the minimum in $\alpha > 0$. These three different scenarios are illustrated in Figure 8 also showing the underlying observations that would lead to them. Figure 1 shows the distribution of $\alpha$-values for two very different optimization trajectories.

**Noisy observations:**   In order to build an approximation for the loss function in the update direction, we leverage the four observations of the function (and its derivative) that are available in each iteration. Due to the stochasticity of deep learning optimization, we also take into account the noise-level of all observations by estimating them. The first two observations are the mini-batch training losses $\mathcal{L}_{\mathcal{B}_t}(\boldsymbol{\theta}_t), \mathcal{L}_{\mathcal{B}_{t+1}}(\boldsymbol{\theta}_{t+1})$ at point $\boldsymbol{\theta}_t$ and $\boldsymbol{\theta}_{t+1}$, which are computed in every standard training loop. The mini-batch losses are averages over individual losses,

$$\mathcal{L}_{\mathcal{B}_t}(\boldsymbol{\theta}_t) = \mathbb{E}_{\mathcal{B}_t}\left[\ell(\boldsymbol{\theta}_t)\right] = \frac{1}{|\mathcal{B}_t|}\sum_{n\in\mathcal{B}_t}\ell_n(\boldsymbol{\theta}_t)\,,$$

$$\mathcal{L}_{\mathcal{B}_{t+1}}(\boldsymbol{\theta}_{t+1}) = \mathbb{E}_{\mathcal{B}_{t+1}}\left[\ell(\boldsymbol{\theta}_{t+1})\right] = \frac{1}{|\mathcal{B}_{t+1}|}\sum_{n\in\mathcal{B}_{t+1}}\ell_n(\boldsymbol{\theta}_{t+1})\,,$$

and using these individual losses, we can also compute the variances to estimate the noise-level of our loss observation,

$$\mathrm{Var}_{\mathcal{B}_t}\left[\ell(\boldsymbol{\theta}_t)\right] = \left(\frac{1}{B_t}\sum_{n\in\mathcal{B}_t}\ell_n(\boldsymbol{\theta}_t)^2\right) - \left(\frac{1}{B_t}\sum_{n\in\mathcal{B}_t}\ell_n(\boldsymbol{\theta}_t)\right)^2\,,$$

$$\mathrm{Var}_{\mathcal{B}_{t+1}}\left[\ell(\boldsymbol{\theta}_{t+1})\right] = \left(\frac{1}{|\mathcal{B}_{t+1}|}\sum_{n\in\mathcal{B}_{t+1}}\ell_n(\boldsymbol{\theta}_{t+1})^2\right) - \left(\frac{1}{|\mathcal{B}_{t+1}|}\sum_{n\in\mathcal{B}_{t+1}}\ell_n(\boldsymbol{\theta}_{t+1})\right)^2\,.$$

Similarly, we proceed with the slope in the update direction. To compute the slope of the loss function in the direction of the optimizer's update $\boldsymbol{s}_t$, we project the current gradient along this update direction

$$\mathbb{E}_{\mathcal{B}_t}\left[\frac{\boldsymbol{s}_t^\top \boldsymbol{g}(\boldsymbol{\theta}_t)}{\|\boldsymbol{s}_t\|^2}\right] = \frac{1}{|\mathcal{B}_t|}\sum_{n \in \mathcal{B}_t}\frac{\boldsymbol{s}_t^\top \boldsymbol{g}_n(\boldsymbol{\theta}_t)}{\|\boldsymbol{s}_t\|^2},$$

$$\mathbb{E}_{\mathcal{B}_{t+1}}\left[\frac{\boldsymbol{s}_t^\top \boldsymbol{g}(\boldsymbol{\theta}_{t+1})}{\|\boldsymbol{s}_t\|^2}\right] = \frac{1}{|\mathcal{B}_{t+1}|}\sum_{n \in \mathcal{B}_{t+1}}\frac{\boldsymbol{s}_t^\top \boldsymbol{g}_n(\boldsymbol{\theta}_{t+1})}{\|\boldsymbol{s}_t\|^2}.$$

Just like before, we can also compute the variance of this slope, by leveraging individual gradients,

$$\text{Var}_{\mathcal{B}_t}\left[\frac{\boldsymbol{s}_t^\top \boldsymbol{g}(\boldsymbol{\theta}_t)}{\|\boldsymbol{s}_t\|^2}\right] = \frac{1}{|\mathcal{B}_t|}\sum_{n \in B_t}\left(\frac{\boldsymbol{s}_t^\top \boldsymbol{g}_n(\boldsymbol{\theta}_t)}{\|\boldsymbol{s}_t\|^2}\right)^2 - \left(\frac{1}{|\mathcal{B}_t|}\sum_{n \in \mathcal{B}_t}\frac{\boldsymbol{s}_t^\top \boldsymbol{g}_n(\boldsymbol{\theta}_t)}{\|\boldsymbol{s}_t\|^2}\right)^2,$$

$$\text{Var}_{\mathcal{B}_{t+1}}\left[\frac{\boldsymbol{s}_t^\top \boldsymbol{g}(\boldsymbol{\theta}_{t+1})}{\|\boldsymbol{s}_t\|^2}\right] = \frac{1}{|\mathcal{B}_{t+1}|}\sum_{n \in \mathcal{B}_{t+1}}\left(\frac{\boldsymbol{s}_t^\top \boldsymbol{g}_n(\boldsymbol{\theta}_{t+1})}{\|\boldsymbol{s}_t\|^2}\right)^2 - \left(\frac{1}{|\mathcal{B}_{t+1}|}\sum_{n \in \mathcal{B}_{t+1}}\frac{\boldsymbol{s}_t^\top \boldsymbol{g}_n(\boldsymbol{\theta}_{t+1})}{\|\boldsymbol{s}_t\|^2}\right)^2.$$

**Quadratic fit & normalization:** Using our (noisy) observations, we are now ready to build an approximation for the loss as a function of the step size, which we will denote as $f(\tau)$. We assume a quadratic function for $f$, which follows recent reports for the loss landscape of neural networks [49], i.e. a function $f(\tau) = w_0 + w_1\tau + w_2\tau^2$ parameterized by $\boldsymbol{w} \in \mathbb{R}^3$. We further assume a Gaussian likelihood of the form

$$p\left(\tilde{\boldsymbol{f}}|\boldsymbol{w}, \boldsymbol{\Phi}\right) = \mathcal{N}\left(\tilde{\boldsymbol{f}}; \boldsymbol{\Phi}^\top \boldsymbol{w}, \boldsymbol{\Lambda}\right) \tag{8}$$

for observations $\tilde{\boldsymbol{f}}$ of the loss and its slope. The observation matrix $\boldsymbol{\Phi}$ and the noise matrix of the observations $\boldsymbol{\Lambda}$ are

$$\boldsymbol{\Phi} = \begin{pmatrix} 1 & 1 & 0 & 0 \\ \tau_1 & \tau_2 & 1 & 1 \\ \tau_1^2 & \tau_2^2 & 2\tau_1 & 2\tau_2 \end{pmatrix}, \qquad \boldsymbol{\Lambda} = \begin{pmatrix} \sigma_{\tilde{f}_1} & 0 & 0 & 0 \\ 0 & \sigma_{\tilde{f}_2} & 0 & 0 \\ 0 & 0 & \sigma_{\tilde{f}_1'} & 0 \\ 0 & 0 & 0 & \sigma_{\tilde{f}_2'} \end{pmatrix},$$

where $\tau$ denotes the position and $\sigma$ denotes the noise-level estimate of the observation. The maximum likelihood solution of Equation (8) for the parameters of our quadratic fit is given by

$$\boldsymbol{w} = \left(\boldsymbol{\Phi}\boldsymbol{\Lambda}^{-1}\boldsymbol{\Phi}^\top\right)^{-1}\boldsymbol{\Phi}\boldsymbol{\Lambda}^{-1}\tilde{\boldsymbol{f}}. \tag{9}$$

Once we have the quadratic fit of the univariate loss function in the update direction, we normalize the scales such that the resulting $\alpha$-value expresses the effective step taken by the optimizer sketched in Figure 8.

**Usage:** The $\alpha$-quantity is related to recent line search approaches [28; 45]. However, instead of searching for an acceptable step by repeated attempts, we instead report the effect of the current step size selection. This could, for example, be used to disentangle the two optimization runs in Figure 1. Additionally, this information could also be used to automatically adapt the learning rate during the training process. But, as discussed in Section 3.3, it isn't trivial what the "correct" decision is, as it might depend on the optimization problem, the training phase, and other factors. Having this $\alpha$-quantity can, however, provide more insight into what kind of steps are used in well-tuned runs with traditional optimizers such as SGD.

### C.3 CABS criterion: Coupling adaptive batch sizes with learning rates (`CABS`)

The CABS criterion, proposed by Balles et al. [4], can be used to adapt the mini-batch size during training with SGD. It relies on the gradient noise and approximately optimizes the objective's expected gain per cost. The adaptation rule is (with learning rate $\eta$)

$$|\mathcal{B}| \leftarrow \eta\frac{\text{Tr}(\boldsymbol{\Sigma}_P(\boldsymbol{\theta}))}{\mathcal{L}_P(\boldsymbol{\theta})}, \tag{10}$$

and the practical implementation approximates $\mathcal{L}_P(\boldsymbol{\theta}) \approx \mathcal{L}_\mathcal{B}(\boldsymbol{\theta}), \mathrm{Tr}(\boldsymbol{\Sigma}_P(\boldsymbol{\theta})) \approx \frac{|\mathcal{B}|-1}{|\mathcal{B}|} \mathrm{Tr}(\hat{\boldsymbol{\Sigma}}_\mathcal{B}(\boldsymbol{\theta}))$ (compare equations (10, 22) and first paragraph of Section 4 in [4]). This yields the quantity computed in cockpit's `CABS` instrument,

$$|\mathcal{B}| \leftarrow \eta \frac{\frac{1}{|\mathcal{B}|}\sum_{j=1}^D \sum_{n\in\mathcal{B}} \left[\boldsymbol{g}_n(\boldsymbol{\theta}) - \boldsymbol{g}_\mathcal{B}(\boldsymbol{\theta})\right]_j^2}{\mathcal{L}_\mathcal{B}(\boldsymbol{\theta})} \, . \tag{11}$$

**Usage:** The CABS criterion suggests a batch size which is optimal under certain assumptions. This suggestion can support practitioners in the batch size selection for their deep learning task.

## C.4 Early-stopping criterion for SGD (`EarlyStopping`)

The empirical risk $\mathcal{L}_\mathcal{D}(\boldsymbol{\theta})$, and the mini-batch loss $\mathcal{L}_\mathcal{B}(\boldsymbol{\theta})$ are only estimators of the target objective $\mathcal{L}_P(\boldsymbol{\theta})$. Mahsereci et al. [29] motivate $p(\boldsymbol{g}_{\mathcal{B},\mathcal{D}}(\boldsymbol{\theta}) \mid \boldsymbol{g}_P(\boldsymbol{\theta}) = \boldsymbol{0})$ as a measure for detecting noise in the finite data sets $\mathcal{B}, \mathcal{D}$ due to sampling from $P$. They propose an evidence-based (EB) criterion for early stopping the training procedure based on mini-batch statistics, and model $p(\boldsymbol{g}_\mathcal{B}(\boldsymbol{\theta}))$ with a sampled diagonal variance approximation (compare Equation (7)),

$$p(\boldsymbol{g}_\mathcal{B}(\boldsymbol{\theta})) \approx \prod_{j=1}^D \mathcal{N}\left([\boldsymbol{g}_P(\boldsymbol{\theta})]_j ; \frac{\left[\hat{\boldsymbol{\Sigma}}_\mathcal{B}(\boldsymbol{\theta})\right]_{j,j}}{|\mathcal{B}|}\right) \, . \tag{12}$$

Their SGD stopping criterion is

$$\frac{2}{D}\left[\log p(\boldsymbol{g}_\mathcal{B}(\boldsymbol{\theta})) - \mathbb{E}_{\boldsymbol{g}_\mathcal{B}(\boldsymbol{\theta}) \sim p(\boldsymbol{g}_\mathcal{B}(\boldsymbol{\theta}))}\left[\log p(\boldsymbol{g}_\mathcal{B}(\boldsymbol{\theta}))\right]\right] > 0 \, , \tag{13a}$$

and translates into

$$1 - \frac{|\mathcal{B}|}{D}\sum_{j=1}^D \frac{[\boldsymbol{g}_\mathcal{B}(\boldsymbol{\theta})]_j^2}{\left[\hat{\boldsymbol{\Sigma}}_\mathcal{B}(\boldsymbol{\theta})\right]_{j,j}} > 0 \, , \tag{13b}$$

$$1 - \frac{|\mathcal{B}|}{D}\sum_{d=1}^D \frac{[\boldsymbol{g}_\mathcal{B}(\boldsymbol{\theta})]_d^2}{\frac{1}{|\mathcal{B}|-1}\sum_{n\in\mathcal{B}}[\boldsymbol{g}_n(\boldsymbol{\theta}) - \boldsymbol{g}_\mathcal{B}(\boldsymbol{\theta})]_d^2} > 0 \, , \tag{13c}$$

$$1 - \frac{|\mathcal{B}|(|\mathcal{B}|-1)}{D}\sum_{d=1}^D \frac{[\boldsymbol{g}_\mathcal{B}(\boldsymbol{\theta})]_d^2}{\left(\sum_{n\in\mathcal{B}}[\boldsymbol{g}_n(\boldsymbol{\theta})]_d^2\right) - |\mathcal{B}|[\boldsymbol{g}_\mathcal{B}(\boldsymbol{\theta})]_d^2} > 0 \, . \tag{13d}$$

COCKPIT's `EarlyStopping` quantity computes the left-hand side of Equation (13d).

**Usage:** The `EarlyStopping` quantity of COCKPIT can inform the practitioner that training is about to be completed and the model might be at risk of overfitting.

## C.5 Individual gradient element histograms (`GradHist1d`, `GradHist2d`)

For the $|\mathcal{B}| \times D$ individual gradient elements, COCKPIT's `GradHist1d` instrument displays a histogram of

$$\{\boldsymbol{g}_n(\boldsymbol{\theta}_j)\}_{n\in\mathcal{B}, j=1,\dots,D} \, . \tag{14}$$

COCKPIT's `GradHist2d` instrument displays a two-dimensional histogram of the $|\mathcal{B}| \times D$ tuples

$$\{(\boldsymbol{\theta}_j, \boldsymbol{g}_n(\boldsymbol{\theta}_j))\}_{n\in\mathcal{B}, j=1,\dots,D} \tag{15}$$

and the marginalized one-dimensional histograms over the parameter and gradient axes.

**Usage:** Sections 3.1 and 3.2 provide use cases (identifying data pre-processing issues and vanishing gradients) for both the gradient histogram as well as its layer-wise extension.

## C.6  Gradient tests (`NormTest, InnerTest, OrthoTest`)

Bollapragada et al. [7] and Byrd et al. [9] propose batch size adaptation schemes based on the gradient noise. They formulate geometric constraints between population and mini-batch gradient and accessible approximations that can be probed to decide whether the mini-batch size should be increased. Because mini-batches are i.i.d. from $P$, it holds that

$$\mathbb{E}\left[g_\mathcal{B}(\boldsymbol{\theta})\right] = g_P(\boldsymbol{\theta}), \tag{16a}$$

$$\mathbb{E}\left[g_\mathcal{B}(\boldsymbol{\theta})^\top g_P(\boldsymbol{\theta})\right] = \|g_P(\boldsymbol{\theta})\|^2. \tag{16b}$$

The above works propose enforcing other weaker similarity in expectation during optimization. These geometric constraints reduce to basic vector geometry (see Figure 9 (a) for an overview of the relevant vectors). We recall their formulation here for consistency and derive the practical versions, which can be computed from training observables and are used in COCKPIT (consult Figure 9 (b) for the visualization).

(a)            (b)

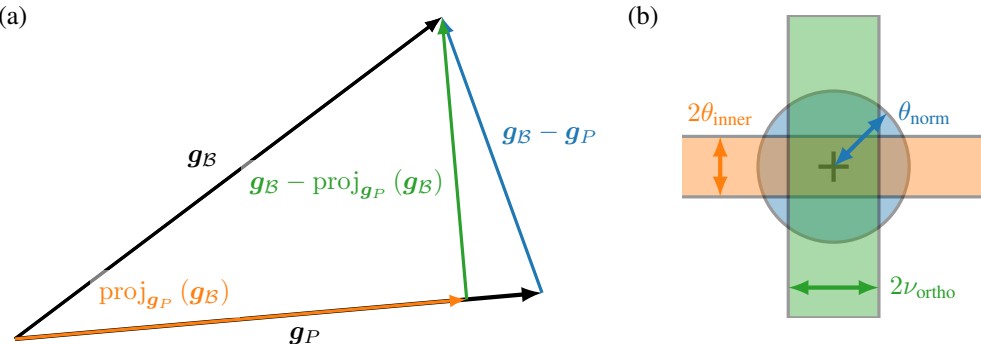

Figure 9: **Conceptual sketch for gradient tests.** *(a)* Relevant vectors to formulate the geometric constraints between population and mini-batch gradient probed by the gradient tests. *(b)* Gradient test visualization in COCKPIT.

**Usage:** All three gradient tests describe the noise level of the gradients. Bollapragada et al. [7] and Byrd et al. [9] adapt the batch size so that the proposed geometric constraints are fulfilled. Practitioners can use the combined gradient test plot, i.e. top center plot in Figure 2, to monitor gradient noise during training and adjust hyperparameters such as the batch size.

### C.6.1  Norm test (`NormTest`)

The norm test [9] constrains the residual norm $\|g_\mathcal{B}(\boldsymbol{\theta}) - g_P(\boldsymbol{\theta})\|$, rescaled by $\|g_P(\boldsymbol{\theta})\|$. This gives rise to a standardized ball of radius $\theta_\text{norm} \in (0, \infty)$ around the population gradient, where the mini-batch gradient should reside. Byrd et al. [9] set $\theta_\text{norm} = 0.9$ in their experiments and increase the batch size if (in the practical version, see below) the following constraint is not fulfilled

$$\mathbb{E}\left[\frac{\|g_\mathcal{B}(\boldsymbol{\theta}) - g_P(\boldsymbol{\theta})\|^2}{\|g_P(\boldsymbol{\theta})\|^2}\right] \leq \theta_\text{norm}^2. \tag{17a}$$

Instead of taking the expectation over mini-batches, Byrd et al. [9] note that the above will be satisfied if

$$\frac{1}{|\mathcal{B}|}\mathbb{E}\left[\frac{\|g_n(\boldsymbol{\theta}) - g_P(\boldsymbol{\theta})\|^2}{\|g_P(\boldsymbol{\theta})\|^2}\right] \leq \theta_\text{norm}^2. \tag{17b}$$

They propose a practical form of this test,

$$\frac{1}{|\mathcal{B}|(|\mathcal{B}|-1)}\frac{\sum_{n\in\mathcal{B}}\|g_n(\boldsymbol{\theta}) - g_\mathcal{B}(\boldsymbol{\theta})\|^2}{\|g_\mathcal{B}(\boldsymbol{\theta})\|^2} \leq \theta_\text{norm}^2, \tag{18a}$$

which can be computed from mini-batch statistics. Rearranging

$$\sum_{n \in \mathcal{B}} \|\boldsymbol{g}_n(\boldsymbol{\theta}) - \boldsymbol{g}_{\mathcal{B}}(\boldsymbol{\theta})\|^2 = \left( \sum_{n \in \mathcal{B}} \|\boldsymbol{g}_n(\boldsymbol{\theta})\|^2 \right) - |\mathcal{B}| \, \|\boldsymbol{g}_{\mathcal{B}}(\boldsymbol{\theta})\|^2 \,, \tag{18b}$$

we arrive at

$$\frac{1}{|\mathcal{B}|(|\mathcal{B}| - 1)} \left[ \frac{\sum_{n \in \mathcal{B}} \|\boldsymbol{g}_n(\boldsymbol{\theta})\|^2}{\|\boldsymbol{g}_{\mathcal{B}}(\boldsymbol{\theta})\|^2} - |\mathcal{B}| \right] \leq \theta_{\text{norm}}^2 \tag{18c}$$

that leverages the norm of both the mini-batch and the individual gradients, which can be aggregated over parameters during a backward pass. COCKPIT's `NormTest` corresponds to the maximum radius $\theta_{\text{norm}}$ for which the above inequality holds.

### C.6.2 Inner product test (`InnerTest`)

The inner product test [7] constrains the projection of $\boldsymbol{g}_{\mathcal{B}}(\boldsymbol{\theta})$ onto $\boldsymbol{g}_P(\boldsymbol{\theta})$ (compare Figure 9 (a)),

$$\text{proj}_{\boldsymbol{g}_P(\boldsymbol{\theta})} (\boldsymbol{g}_{\mathcal{B}}(\boldsymbol{\theta})) = \frac{\boldsymbol{g}_{\mathcal{B}}(\boldsymbol{\theta})^\top \boldsymbol{g}_P(\boldsymbol{\theta})}{\|\boldsymbol{g}_P(\boldsymbol{\theta})\|^2} \boldsymbol{g}_P(\boldsymbol{\theta}) \,, \tag{19}$$

rescaled by $\|\boldsymbol{g}_P(\boldsymbol{\theta})\|$. This restricts the mini-batch gradient to reside in a standardized band of relative width $\theta_{\text{inner}} \in (0, \infty)$ around the population risk gradient. Bollapragada et al. [7] use $\theta_{\text{inner}} = 0.9$ (in the practical version, see below) to adapt the batch size if the parallel component's variance does not satisfy the condition

$$\text{Var} \left( \frac{\boldsymbol{g}_{\mathcal{B}}(\boldsymbol{\theta})^\top \boldsymbol{g}_P(\boldsymbol{\theta})}{\|\boldsymbol{g}_P(\boldsymbol{\theta})\|^2} \right) = \mathbb{E} \left[ \left( \frac{\boldsymbol{g}_{\mathcal{B}}(\boldsymbol{\theta})^\top \boldsymbol{g}_P(\boldsymbol{\theta})}{\|\boldsymbol{g}_P(\boldsymbol{\theta})\|^2} - 1 \right)^2 \right] \leq \theta_{\text{inner}}^2 \tag{20a}$$

(note that by Equation (16) we have $\mathbb{E} \left[ \frac{\boldsymbol{g}_{\mathcal{B}}(\boldsymbol{\theta})^\top \boldsymbol{g}_P(\boldsymbol{\theta})}{\|\boldsymbol{g}_P(\boldsymbol{\theta})\|^2} \right] = 1$). Bollapragada et al. [7] bound Equation (20a) by the individual gradient variance,

$$\frac{1}{|\mathcal{B}|} \text{Var} \left( \frac{\boldsymbol{g}_n(\boldsymbol{\theta})^\top \boldsymbol{g}_P(\boldsymbol{\theta})}{\|\boldsymbol{g}_P(\boldsymbol{\theta})\|^2} \right) = \frac{1}{|\mathcal{B}|} \mathbb{E} \left[ \left( \frac{\boldsymbol{g}_n(\boldsymbol{\theta})^\top \boldsymbol{g}_P(\boldsymbol{\theta})}{\|\boldsymbol{g}_P(\boldsymbol{\theta})\|^2} - 1 \right)^2 \right] \leq \theta_{\text{inner}}^2 \,. \tag{20b}$$

They then propose a practical form of Equation (20b), which uses the mini-batch sample variance,

$$\frac{1}{|\mathcal{B}|} \text{Var} \left( \frac{\boldsymbol{g}_n(\boldsymbol{\theta})^\top \boldsymbol{g}_{\mathcal{B}}(\boldsymbol{\theta})}{\|\boldsymbol{g}_{\mathcal{B}}(\boldsymbol{\theta})\|^2} \right) = \frac{1}{|\mathcal{B}|(|\mathcal{B}| - 1)} \left[ \sum_{n \in \mathcal{B}} \left( \frac{\boldsymbol{g}_n(\boldsymbol{\theta})^\top \boldsymbol{g}_{\mathcal{B}}(\boldsymbol{\theta})}{\|\boldsymbol{g}_{\mathcal{B}}(\boldsymbol{\theta})\|^2} - 1 \right)^2 \right] \leq \theta_{\text{inner}}^2 \,. \tag{21a}$$

Expanding

$$\sum_{n \in \mathcal{B}} \left( \frac{\boldsymbol{g}_n(\boldsymbol{\theta})^\top \boldsymbol{g}_{\mathcal{B}}(\boldsymbol{\theta})}{\|\boldsymbol{g}_{\mathcal{B}}(\boldsymbol{\theta})\|^2} - 1 \right)^2 = \frac{\sum_{n \in \mathcal{B}} \left( \boldsymbol{g}_n(\boldsymbol{\theta})^\top \boldsymbol{g}_{\mathcal{B}}(\boldsymbol{\theta}) \right)^2}{\|\boldsymbol{g}_{\mathcal{B}}(\boldsymbol{\theta})\|^4} - |\mathcal{B}| \tag{21b}$$

and inserting Equation (21b) into Equation (21a) yields

$$\frac{1}{|\mathcal{B}|(|\mathcal{B}| - 1)} \left[ \frac{\sum_{n \in \mathcal{B}} \left( \boldsymbol{g}_n(\boldsymbol{\theta})^\top \boldsymbol{g}_{\mathcal{B}}(\boldsymbol{\theta}) \right)^2}{\|\boldsymbol{g}_{\mathcal{B}}(\boldsymbol{\theta})\|^4} - |\mathcal{B}| \right] \leq \theta_{\text{inner}}^2 \,. \tag{21c}$$

It relies on pairwise scalar products between individual gradients, which can be aggregated over layers during backpropagation. COCKPIT's `InnerTest` quantity computes the maximum band width $\theta_{\text{inner}}$ that satisfies Equation (21c).

### C.6.3 Orthogonality test (`OrthoTest`)

In contrast to the inner product test (Appendix C.6.2) which constrains the projection (Equation (19)), the orthogonality test [7] constrains the orthogonal part (see Figure 9 (a))

$$\boldsymbol{g}_{\mathcal{B}}(\boldsymbol{\theta}) - \mathrm{proj}_{\boldsymbol{g}_P(\boldsymbol{\theta})}\left(\boldsymbol{g}_{\mathcal{B}}(\boldsymbol{\theta})\right) , \tag{22}$$

rescaled by $\|\boldsymbol{g}_P(\boldsymbol{\theta})\|$. This restricts the mini-batch gradient to a standardized band of relative width $\nu_{\mathrm{ortho}} \in (0, \infty)$ parallel to the population gradient. Bollapragada et al. [7] use $\nu = \tan(80°) \approx 5.84$ (in the practical version, see below) to adapt the batch size if the following condition is violated,

$$\mathbb{E}\left[\left\|\frac{\boldsymbol{g}_{\mathcal{B}}(\boldsymbol{\theta}) - \mathrm{proj}_{\boldsymbol{g}_P(\boldsymbol{\theta})}\left(\boldsymbol{g}_{\mathcal{B}}(\boldsymbol{\theta})\right)}{\|\boldsymbol{g}_P(\boldsymbol{\theta})\|}\right\|^2\right] \leq \nu_{\mathrm{ortho}}^2 . \tag{23a}$$

Expanding the norm, and inserting Equation (19), this simplifies to

$$\mathbb{E}\left[\left\|\frac{\boldsymbol{g}_{\mathcal{B}}(\boldsymbol{\theta})}{\|\boldsymbol{g}_P(\boldsymbol{\theta})\|} - \frac{\boldsymbol{g}_{\mathcal{B}}(\boldsymbol{\theta})^\top \boldsymbol{g}_P(\boldsymbol{\theta})}{\|\boldsymbol{g}_P(\boldsymbol{\theta})\|^2}\frac{\boldsymbol{g}_P(\boldsymbol{\theta})}{\|\boldsymbol{g}_P(\boldsymbol{\theta})\|}\right\|^2\right] \leq \nu_{\mathrm{ortho}}^2 ,$$

$$\mathbb{E}\left[\frac{\|\boldsymbol{g}_{\mathcal{B}}(\boldsymbol{\theta})\|^2}{\|\boldsymbol{g}_P(\boldsymbol{\theta})\|^2} - \frac{\left(\boldsymbol{g}_{\mathcal{B}}(\boldsymbol{\theta})^\top \boldsymbol{g}_P(\boldsymbol{\theta})\right)^2}{\|\boldsymbol{g}_P(\boldsymbol{\theta})\|^4}\right] \leq \nu_{\mathrm{ortho}}^2 . \tag{23b}$$

Bollapragada et al. [7] bound this inequality using individual gradients instead,

$$\frac{1}{|\mathcal{B}|}\mathbb{E}\left[\left\|\frac{\boldsymbol{g}_n(\boldsymbol{\theta})}{\|\boldsymbol{g}_P(\boldsymbol{\theta})\|^2} - \frac{\boldsymbol{g}_n(\boldsymbol{\theta})^\top \boldsymbol{g}_P(\boldsymbol{\theta})}{\|\boldsymbol{g}_P(\boldsymbol{\theta})\|^2}\frac{\boldsymbol{g}_P(\boldsymbol{\theta})}{\|\boldsymbol{g}_P(\boldsymbol{\theta})\|}\right\|^2\right] \leq \nu_{\mathrm{ortho}}^2 . \tag{23c}$$

They propose the practical form

$$\frac{1}{|\mathcal{B}|(|\mathcal{B}| - 1)}\mathbb{E}\left[\left\|\frac{\boldsymbol{g}_n(\boldsymbol{\theta})}{\|\boldsymbol{g}_{\mathcal{B}}(\boldsymbol{\theta})\|} - \frac{\boldsymbol{g}_n(\boldsymbol{\theta})^\top \boldsymbol{g}_{\mathcal{B}}(\boldsymbol{\theta})}{\|\boldsymbol{g}_{\mathcal{B}}(\boldsymbol{\theta})\|^2}\frac{\boldsymbol{g}_{\mathcal{B}}(\boldsymbol{\theta})}{\|\boldsymbol{g}_{\mathcal{B}}(\boldsymbol{\theta})\|}\right\|^2\right] \leq \nu_{\mathrm{ortho}}^2 , \tag{24a}$$

which simplifies to

$$\frac{1}{|\mathcal{B}|(|\mathcal{B}| - 1)}\sum_{n \in \mathcal{B}}\left(\frac{\|\boldsymbol{g}_n(\boldsymbol{\theta})\|^2}{\|\boldsymbol{g}_{\mathcal{B}}(\boldsymbol{\theta})\|^2} - \frac{\left(\boldsymbol{g}_n(\boldsymbol{\theta})^\top \boldsymbol{g}_{\mathcal{B}}(\boldsymbol{\theta})\right)^2}{\|\boldsymbol{g}_{\mathcal{B}}(\boldsymbol{\theta})\|^4}\right) \leq \nu_{\mathrm{ortho}}^2 . \tag{24b}$$

It relies on pairwise scalar products between individual gradients which can be aggregated over layers during a backward pass. COCKPIT's `OrthTest` quantity computes the maximum band width $\nu_{\mathrm{ortho}}$ which satisfies Equation (24b).

**Relation to acute angle test:** Recently, a novel "acute angle test" was proposed by Bahamou & Goldfarb [3]. While the theoretical constraint between $\boldsymbol{g}_{\mathcal{B}}(\boldsymbol{\theta})$ and $\boldsymbol{g}_P(\boldsymbol{\theta})$ differs from the orthogonality test, the practical versions coincide. Hence, we do not incorporate the acute angle here.

### C.7 Hessian maximum eigenvalue (`HessMaxEV`)

The Hessian's maximum eigenvalue $\lambda_{\max}(\boldsymbol{H}_{\mathcal{B}}(\boldsymbol{\theta}))$ is computed with an iterative eigensolver from Hessian-vector products through PYTORCH's automatic differentiation [34]. Like Yao et al. [50], we employ power iterations with similar default stopping parameters (stop after at most 100 iterations, or if the iterate does converged with a relative and absolute tolerance of $10^{-3}, 10^{-6}$, respectively) to compute $\lambda_{\max}(\boldsymbol{H}_{\mathcal{B}}(\boldsymbol{\theta}))$ through the `HessMaxEV` quantity in COCKPIT.

In principle, more sophisticated eigensolvers (for example Arnoldi's method) could be applied to converge in fewer iterations or compute eigenvalues other than the leading ones. Warsa et al. [46] empirically demonstrate that the FLOP ratio between power iteration and implicitly restarted Arnoldi method can reach values larger than 100. While we can use such a beneficial method on a CPU through `scipy.sparse.linalg.eigsh` we are restricted to the GPU-compatible power iteration for GPU training. We expect that extending the support of popular machine learning libraries like PYTORCH for such iterative eigensolvers on GPUs can help to save computation time.

$$\lambda_{\max}(\boldsymbol{H}_{\mathcal{B}}(\boldsymbol{\theta})) = \max_{\|\boldsymbol{v}\|=1}\|\boldsymbol{H}_{\mathcal{B}}(\boldsymbol{\theta})\boldsymbol{v}\| = \max_{\boldsymbol{v}\in\mathbb{R}^D}\frac{\boldsymbol{v}^\top \boldsymbol{H}_{\mathcal{B}}(\boldsymbol{\theta})\boldsymbol{v}}{\boldsymbol{v}^\top \boldsymbol{v}} . \tag{25}$$

**Usage:** The Hessian's maximum eigenvalue describes the loss surface's sharpest direction and thus provides an understanding of the current loss landscape. Additionally, in convex optimization, the largest Hessian eigenvalue crucially determines the appropriate step size [38]. In Section 4, we can observe that although training seems stuck in the very first few iterations progress is visible when looking at the maximum Hessian eigenvalue.

### C.8 Hessian trace (`HessTrace`)

In comparison to Yao et al. [50], who leverage Hessian-vector products [34] to estimate the Hessian trace, we compute the exact value $\mathrm{Tr}(\boldsymbol{H}_{\mathcal{B}}(\boldsymbol{\theta}))$ with the `HessTrace` quantity in COCKPIT by aggregating the output of BACKPACK's `DiagHessian` extension, which computes the diagonal entries of $\boldsymbol{H}_{\mathcal{B}}(\boldsymbol{\theta})$. Alternatively, the trace can also be estimated from the generalized Gauss-Newton matrix, or an MC-sampled approximation thereof.

**Usage:** The Hessian trace equals the sum of the eigenvalues and thus provides a notion of "average curvature" of the current loss landscape. It has long been theorized and discussed that curvature and generalization performance may be linked [21, e.g.].

### C.9 Takeuchi Information Criterion (TIC) (`TICDiag, TICTrace`)

Recent work by Thomas et al. [43] suggests that optimizer convergence speed and generalization is mainly influenced by curvature and gradient noise; and hence their interaction is crucial to understand the generalization and optimization behavior of deep neural networks. They reinvestigate the Takeuchi Information criterion [42], an estimator for the generalization gap in overparameterized maximum likelihood estimation. At a local minimum $\boldsymbol{\theta}^{\star}$, the generalization gap is estimated by the TIC

$$\frac{1}{|\mathcal{D}|} \mathrm{Tr}\left(\boldsymbol{H}_P(\boldsymbol{\theta}^{\star})^{-1} \boldsymbol{C}_P(\boldsymbol{\theta}^{\star})\right), \tag{26}$$

where $\boldsymbol{H}_P(\boldsymbol{\theta}^{\star})$ is the population Hessian and $\boldsymbol{C}_P(\boldsymbol{\theta}^{\star})$ is the gradient's uncentered second moment,

$$\boldsymbol{C}_P(\boldsymbol{\theta}^{\star}) = \int \nabla_{\boldsymbol{\theta}}\ell(f(\boldsymbol{\theta}^{\star}, \boldsymbol{x}), \boldsymbol{y}) \left(\nabla_{\boldsymbol{\theta}}\ell(f(\boldsymbol{\theta}^{\star}, \boldsymbol{x}), \boldsymbol{y})\right)^{\top} P(\boldsymbol{x}, \boldsymbol{y}) \, d\boldsymbol{x} \, d\boldsymbol{y}.$$

Both matrices are inaccessible in practice. In their experiments, Thomas et al. [43] propose the approximation $\mathrm{Tr}(\boldsymbol{C})/\mathrm{Tr}(\boldsymbol{H})$ for $\mathrm{Tr}(\boldsymbol{H}^{-1}\boldsymbol{C})$. They also replace the Hessian by the Fisher as it is easier to compute. With these practical simplifications, they investigate the TIC of trained neural networks where the curvature and noise matrix are evaluated on a large data set.

The TIC provided in COCKPIT differs from this setting, since by design we want to observe quantities during training, while avoiding additional model predictions. Also, BACKPACK provides access to the Hessian; hence we don't need to use the Fisher. We propose the following two approximations of the TIC from a mini-batch:

- `TICTrace`: Uses the approximation of Thomas et al. [43] which replaces the matrix-product trace by the product of traces,

$$\frac{\mathrm{Tr}\left(\boldsymbol{C}_{\mathcal{B}}(\boldsymbol{\theta})\right)}{\mathrm{Tr}\left(\boldsymbol{H}_{\mathcal{B}}(\boldsymbol{\theta})\right)} = \frac{\frac{1}{|\mathcal{B}|}\sum_{n\in\mathcal{B}}\|\boldsymbol{g}_n(\boldsymbol{\theta})\|^2}{\mathrm{Tr}\left(\boldsymbol{H}_{\mathcal{B}}(\boldsymbol{\theta})\right)}. \tag{27}$$

- `TICDiag`: Uses a diagonal approximation of the Hessian, which is cheap to invert,

$$\mathrm{Tr}\left(\mathrm{diag}\left(\boldsymbol{H}_{\mathcal{B}}(\boldsymbol{\theta})\right)^{-1}\boldsymbol{C}_{\mathcal{B}}(\boldsymbol{\theta})\right) = \frac{1}{|\mathcal{B}|}\sum_{j=1}^{D}[\boldsymbol{H}_{\mathcal{B}}(\boldsymbol{\theta})]_{j,j}^{-1}\left[\sum_{n\in\mathcal{B}}\boldsymbol{g}_n(\boldsymbol{\theta})^{\odot 2}\right]_j. \tag{28}$$

**Usage:** The TIC is a proxy for the model's generalization gap, see Thomas et al. [43].

## C.10 Gradient signal-to-noise-ratio (`MeanGSNR`)

The gradient signal-to-noise-ratio $\mathrm{GSNR}([\boldsymbol{\theta}]_j) \in \mathbb{R}$ for a single parameter $[\boldsymbol{\theta}]_j$ is defined as

$$\mathrm{GSNR}([\boldsymbol{\theta}]_j) = \frac{\mathbb{E}_{(\boldsymbol{x},\boldsymbol{y})\sim P}\left[[\nabla_{\boldsymbol{\theta}}\ell(f(\boldsymbol{\theta},\boldsymbol{x}),\boldsymbol{y})]_j\right]^2}{\mathrm{Var}_{(\boldsymbol{x},\boldsymbol{y})\sim P}\left[[\nabla_{\boldsymbol{\theta}}\ell(f(\boldsymbol{\theta},\boldsymbol{x}),\boldsymbol{y})]_j\right]} = \frac{[\boldsymbol{g}_P(\boldsymbol{\theta})]_j^2}{[\boldsymbol{\Sigma}_P(\boldsymbol{\theta})]_{j,j}}. \tag{29}$$

Liu et al. [27] use it to explain generalization properties of models in the early training phase. We apply their estimation to mini-batches,

$$\mathrm{GSNR}([\boldsymbol{\theta}]_j) \approx \frac{[\boldsymbol{g}_\mathcal{B}(\boldsymbol{\theta})]_j^2}{\frac{|\mathcal{B}|-1}{|\mathcal{B}|}\left[\hat{\boldsymbol{\Sigma}}_\mathcal{B}(\boldsymbol{\theta})\right]_{j,j}} = \frac{[\boldsymbol{g}_\mathcal{B}(\boldsymbol{\theta})]_j^2}{\frac{1}{|\mathcal{B}|}\left(\sum_{n\in\mathcal{B}}[\boldsymbol{g}_n(\boldsymbol{\theta})]_j^2\right) - [\boldsymbol{g}_\mathcal{B}(\boldsymbol{\theta})]_j^2}. \tag{30a}$$

Inspired by Liu et al. [27], COCKPIT's `MeanGSNR` computes the average GSNR over all parameters,

$$\frac{1}{D}\sum_{j=1}^{D}\mathrm{GSNR}([\boldsymbol{\theta}]_j). \tag{30b}$$

**Usage:** The GSNR describes the gradient noise level which is influenced, among other things, by the batch size. Using the GSNR, perhaps in combination with the gradient tests or the CABS criterion could provide practitioners a clearer picture of suitable batch sizes for their particular problem. As shown by Liu et al. [27], the GSNR is also linked to generalization of neural networks.

# D Additional experiments

In this section, we present additional experiments and use cases that showcase COCKPIT's utility. Appendix D.1 shows that COCKPIT is able to scale to larger data sets by running the experiment with incorrectly scaled data (see Section 3.1) on IMAGENET instead of CIFAR-10. Appendix D.2 provides another concrete use case similar to Figure 1: detecting regularization during training.

## D.1 Incorrectly scaled data for IMAGENET

We repeat the experiment of Section 3.1 on the IMAGENET [13] data set instead of CIFAR-10. We also use a larger neural network model, switching from 3C3D to VGG16 [40]. This demonstrates that COCKPIT is able to scale to both larger models and data sets. The input size of the images is almost fifty times larger ($224 \times 224$ instead of $32 \times 32$). The model size increased by roughly a factor of 150 (VGG16 for IMAGENET has roughly 138 million parameters, 3C3D has less than a million).

Similar to the example shown in the main text, the gradients are affected by the scaling introduced via the input images, albeit less drastically (see Figure 10). Due to the gradient scaling, default optimization hyperparameters might not work well anymore for the model using the raw input data.

## D.2 Detecting implicit regularization of the optimizer

In non-convex optimization, optimizers can converge to local minima with different properties. Here, we illustrate this by investigating the effect of sub-sampling noise on a simple task from [30; 18].

We generate synthetic data $\mathcal{D} = \{(x_n, y_n) \in \mathbb{R} \times \mathbb{R}\}_{n=1}^{N=100}$ for a regression task with $x \sim \mathcal{N}(0; 1)$ with noisy observations $y = 1.4x + \epsilon$ where $\epsilon \sim \mathcal{N}(0; 1)$. The model is a scalar network with parameters $\boldsymbol{\theta} = (w_1 \quad w_2)^\top \in \mathbb{R}^2$, initialized at $\boldsymbol{\theta}_0 = (0.1 \quad 1.7)^\top$, that produces predictions via $f(\boldsymbol{\theta}, x) = w_2 w_1 x$. We seek to minimize the mean squared error

$$\mathcal{L}_\mathcal{D}(\boldsymbol{\theta}) = \frac{1}{N}\sum_{n=1}^{N}\left(f(\theta, x_n) - y_n\right)^2$$

and compare SGD ($|\mathcal{B}| = 95$) with GD ($|\mathcal{B}| = N = 100$) at a learning rate of 0.1 (see Figure 11).

We observe that the loss of both SGD and GD is almost identical. Using a noisy gradient regularizes the Hessian's maximum eigenvalue though. It decreases in later stages where the loss curve suggests that training has converged. This regularization effect constitutes an important phenomenon that cannot be observed by monitoring only the loss.

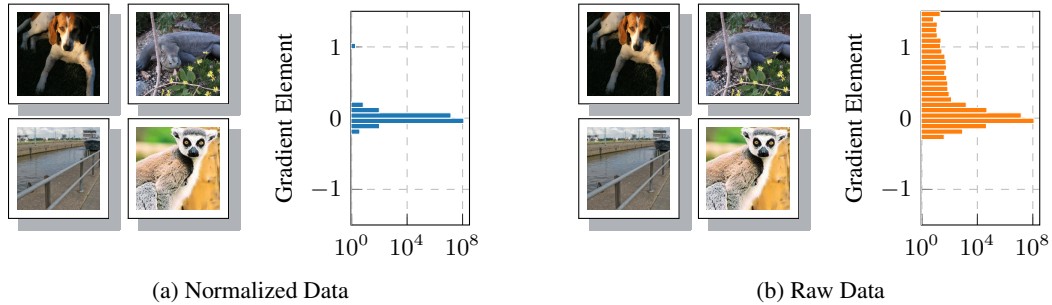

(a) Normalized Data             (b) Raw Data

Figure 10: **Same inputs, different gradients on IMAGENET.** This is structurally the same plot as Figure 3, but using IMAGENET and VGG16. (a) *normalized* ($[0, 1]$) and (b) *raw* ($[0, 255]$) images look identical in auto-scaled front-ends like MATPLOTLIB's imshow. The gradient distribution on the VGG16 model, however, is affected by this scaling.

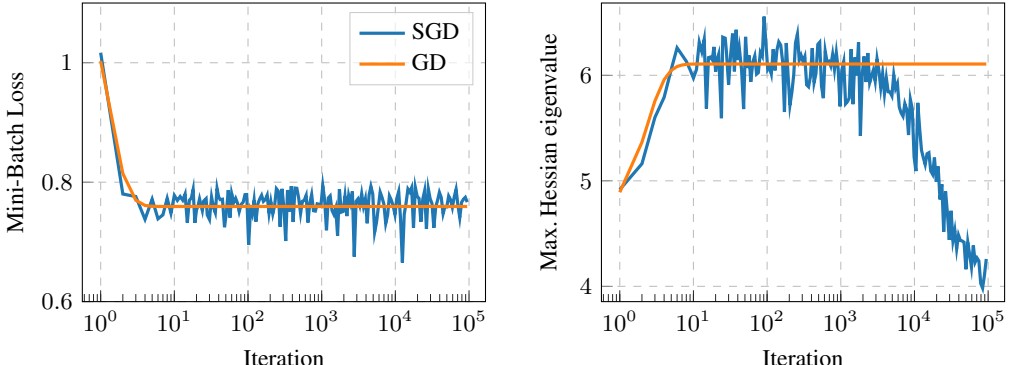

Figure 11: **Observing implicit regularization of the optimizer with COCKPIT** through a comparison of SGD and GD on a synthetic problem inspired by [30; 18] (details in the text). *Left:* The mini-batch loss of both optimizers looks similar. *Right:* Noise due to mini-batching regularizes the Hessian's maximum eigenvalue in stages where the loss suggests that training has converged.

# E   Implementation details and additional benchmarks

In this section, we provide more details about our implementation (Appendix E.1) to access the desired quantities with as little overhead as possible. Additionally, we present more benchmarks for individual instruments (Appendix E.2.1) and COCKPIT configurations (Appendix E.2.2). These are similar but extended versions of the ones presented in Figures 6a and 6b in the main text. Lastly, we benchmark different implementations of computing the two-dimensional gradient histogram (Appendix E.3), identifying a computational bottleneck for its current GPU implementation.

**Hardware details:**   Throughout this paper, we conducted benchmarks on the following setup

- **CPU:** Intel Core i7-8700K CPU @ 3.70 GHz × 12 (32 GB)
- **GPU:** NVIDIA GeForce RTX 2080 Ti (11 GB)

**Test problem details:**   The experiments in this paper rely mostly on optimization problems provided by the DEEPOBS benchmark suite [39]. If not stated otherwise, we use the default training details suggested by DEEPOBS, that are summarized below. For more details see the original paper.

- **Quadratic Deep:** A stochastic quadratic problem with an eigenspectrum similar to what has been reported for neural nets. Default batch size 128, default number of epochs 100.
- **MNIST Log. Reg.:** Multinomial logistic regression on MNIST [26]. Default batch size 128, default number of epochs 50.
- **MNIST MLP:** Multi-layer perceptron neural network on MNIST. Default batch size 128, default number of epochs 100.
- **FASHION-MNIST MLP:** Multi-layer perceptron neural network on FASHION-MNIST [48]. Default batch size 128, default number of epochs 100.
- **FASHION-MNIST 2C2D:** A two convolutional and two dense layered neural network on FASHION-MNIST. Default batch size 128, default number of epochs 100.
- **CIFAR-10 3C3D:** A three convolutional and three dense layered neural network on CIFAR-10 [25]. Default batch size 128, default number of epochs 100.
- **CIFAR-100 ALL-CNN-C:** All Convolutional Neural Network C (ALL-CNN-C [41]) on CIFAR-100 [25]. Default batch size 256, default number of epochs 350.
- **SVHN 3C3D:** A three convolutional and three dense layered neural network on SVHN [32]. Default batch size 128, default number of epochs 100.

## E.1   Hooks & Memory benchmarks

To improve memory consumption, we compact information during the backward pass by adding hooks to the neural network's layers. These are executed after BACKPACK extensions and have access to the quantities computed therein. They compress information to what is requested by a quantity and free the memory occupied by BACKPACK buffers. Such savings primarily depend on the parameter distribution over layers, and are bigger for more balanced architectures (compare Figure 12).

**Example:**   Say, we want to compute a histogram over the $|\mathcal{B}| \times D$ individual gradient elements of a network. Suppose that $|\mathcal{B}| = 128$ and the model is DEEPOBS's CIFAR-10 3C3D test problem with $895,210$ parameters. Given that every parameter is stored in single precision, the model requires $895,210 \times 4$ Bytes $\approx 3.41$ MB. Storing the individual gradients will require $128 \times 895,210 \times 4$ Bytes $\approx 437$ MB (for larger networks this quickly exceeds the available memory as the individual gradients occupy $|\mathcal{B}|$ times the model size). If instead, the layer-wise individual gradients are condensed into histograms of negligible size and immediately freed afterwards during backpropagation, the maximum memory overhead reduces to storing the individual gradients of the largest layer. For our example, the largest layer has $589,824$ parameters, and the associated individual gradients will require $128 \times 589,824 \times 4$ Bytes $\approx 288$ MB, saving roughly $149$ MB of RAM. In practice, we observe these expected savings, see Figure 12c.

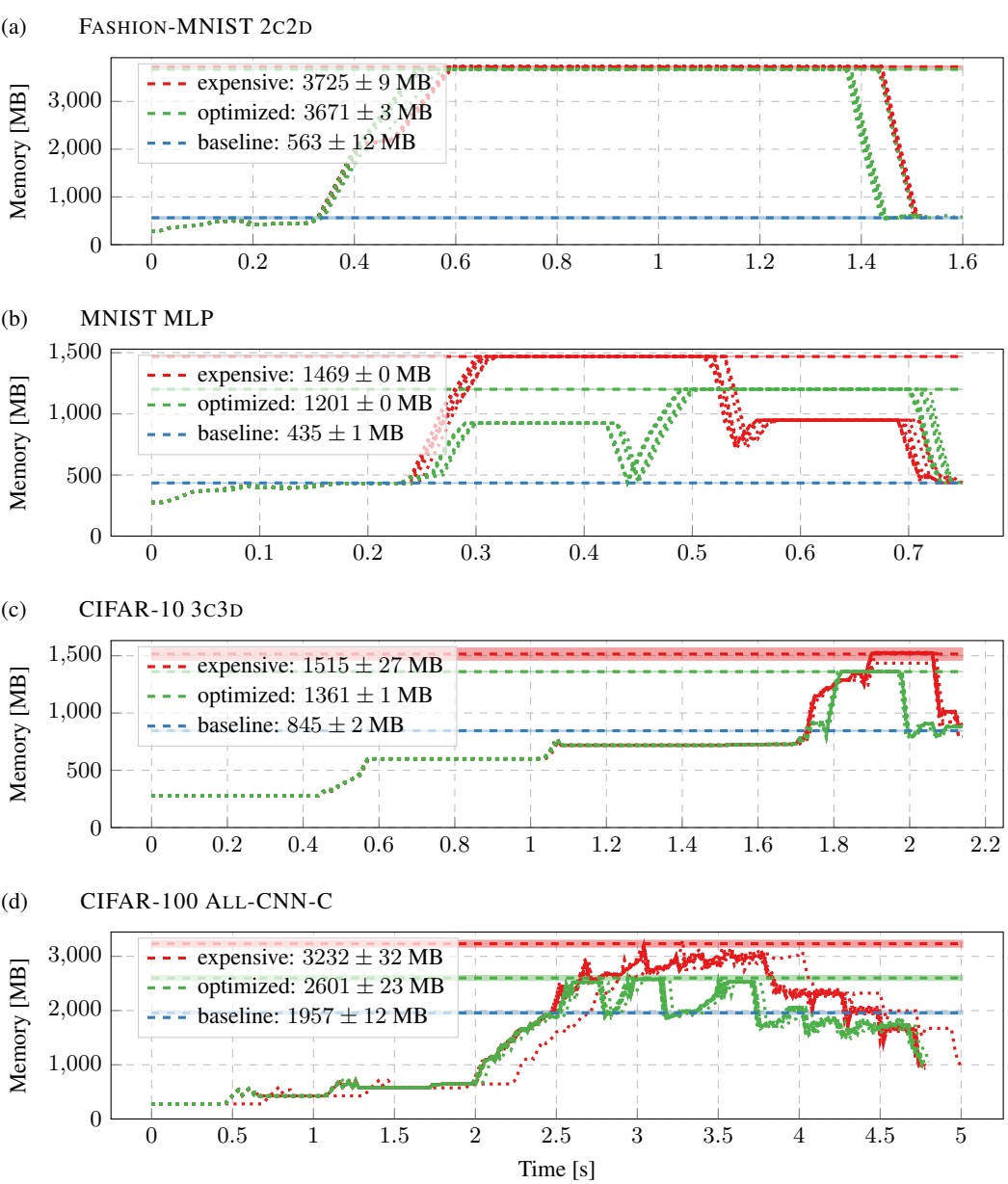

Figure 12: **Memory consumption and savings with hooks** during one forward-backward step on a CPU for different DEEPOBS problems. We compare three settings; i) without COCKPIT (baseline); ii) COCKPIT with GradHist1d with BACKPACK (expensive); iii) COCKPIT with GradHist1d with BACKPACK and additional hooks (optimized). Peak memory consumptions are highlighted by horizontal dashed bars and shown in the legend. Shaded areas, if visible, fill two standard deviations above and below the mean value, all of them result from ten independent runs. Dotted lines indicate individual runs. Our optimized approach allows to free obsolete tensors during backpropagation and thereby reduces memory consumption. From top to bottom: the effect is less pronounced for architectures that concentrate the majority of parameters in a single layer ((a) $3, 274, 634$ total, $3, 211, 264$ largest layer) and increases for more balanced networks ((b) $1, 336, 610$ total, $784, 000$ largest layer, (c): $895, 210$ total, $589, 824$ largest layer).

### E.2 Additional run time benchmarks

#### E.2.1 Individual instrument overhead

To estimate the computational overhead for individual instruments, we run COCKPIT with that instrument for 32 iterations, tracking at every step. Training proceeds with the default batch size specified by the DEEPOBS problem and uses SGD with learning rate $10^{-3}$. We measure the time between iterations 1 and 32, and average for the overhead per step. Every such estimate is repeated over 10 random seeds to obtain mean and error bars as reported in Figure 6a.

Note that this protocol does *not* include initial overhead for setting up data loading and also does *not* include the time for evaluating train/test loss on a larger data set, which is usually done by practitioners. Hence, we even expect the shown overheads to be smaller in a conventional training loop which includes the above steps.

**Individual overhead on GPU versus CPU:**    Figure 13 and Figure 14 show the individual overhead for four different DEEPOBS problems on GPU and CPU, respectively. The left part of Figure 13 (c) corresponds to Figure 6a. Right panels show the expensive quantities, which we omitted in the main text as they were expected to be expensive due to their computational work (HessMaxEV) or bottlenecks in the implementation (GradHist2d, see Appendix E.3 for details). We see that they are in many cases equally or more expensive than computing all other instruments. Another expected feature of the GPU-to-CPU comparison is that parallelism on the CPU is significantly less pronounced. Hence, we observe an increased overhead for all quantities that contain non-linear transformations and contractions of the high-dimensional individual gradients, or require additional backpropagations (curvature).

#### E.2.2 Configuration overhead

For the estimation of different COCKPIT configuration overheads, we use almost the same setting as described above, training for 512 iterations and tracking only every specified interval.

**Configuration overhead on GPU versus CPU:**    Figure 15 and Figure 16 show the configuration overhead for four different DEEPOBS problems. The bottom left part of Figure 15 corresponds to Figure 6b. In general, we observe that increased parallelism can be exploited on a GPU, leading to smaller overheads in comparison to a CPU.

COCKPIT can even scale to significantly larger problems, such as a RESNET-50 on IMAGENET-like data. Figure 17 shows the computational overhead for different tracking intervals on such a large-scale problem. Using the *economy* configuration, we can achieve our self-imposed goal of at most doubling the run time even when tracking every fourth step. More extensive configurations (such as the *full* set) would indeed have almost prohibitively large costs associated. However, these costs could be dramatically reduced when one decides to only inspect a part of the network using COCKPIT. Note, individual gradients are not properly defined when using batch norm, therefore, we replaced these batch norm layers with identity layers when using the RESNET-50.

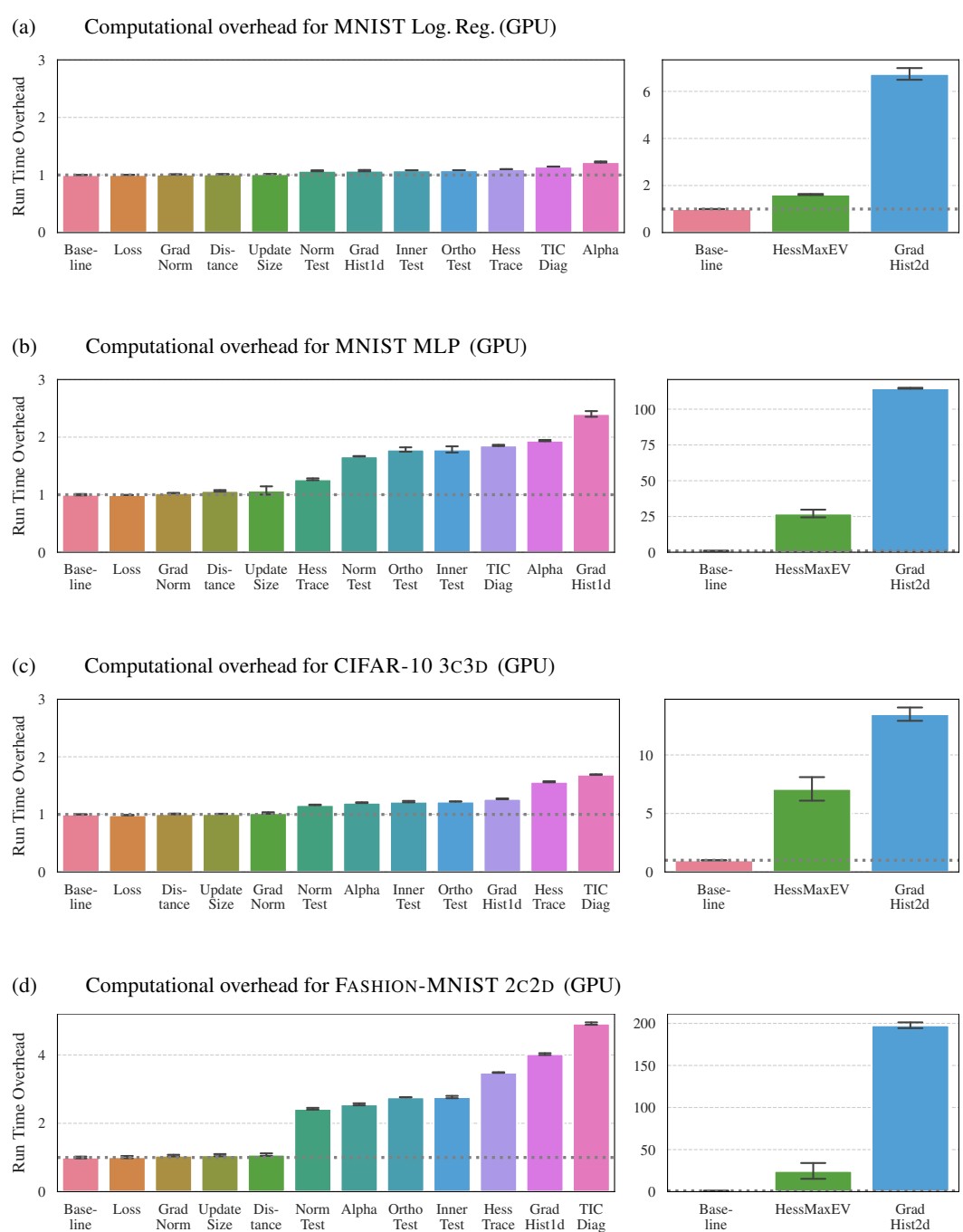

Figure 13: **Individual overhead of COCKPIT's instruments on GPU for four different problems.**
All run times are shown as multiples of the *baseline* without tracking. Expensive quantities are
displayed in separate panels on the right. Experimental details in the text.

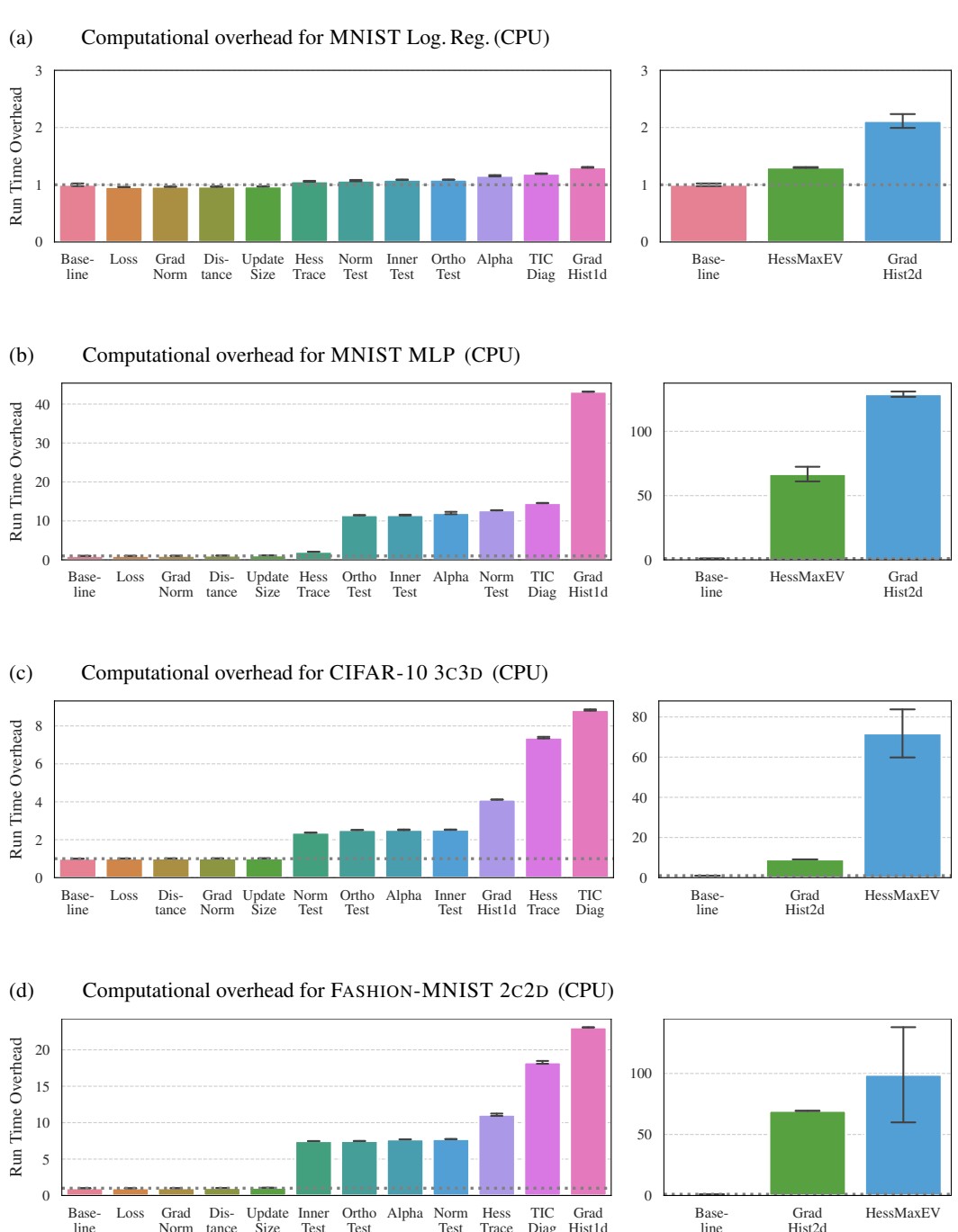

Figure 14: **Individual overhead of COCKPIT's instruments on CPU for four different problems.**
All run times are shown as multiples of the *baseline* without tracking. Expensive quantities are
displayed in separate panels on the right. Experimental details in the text.

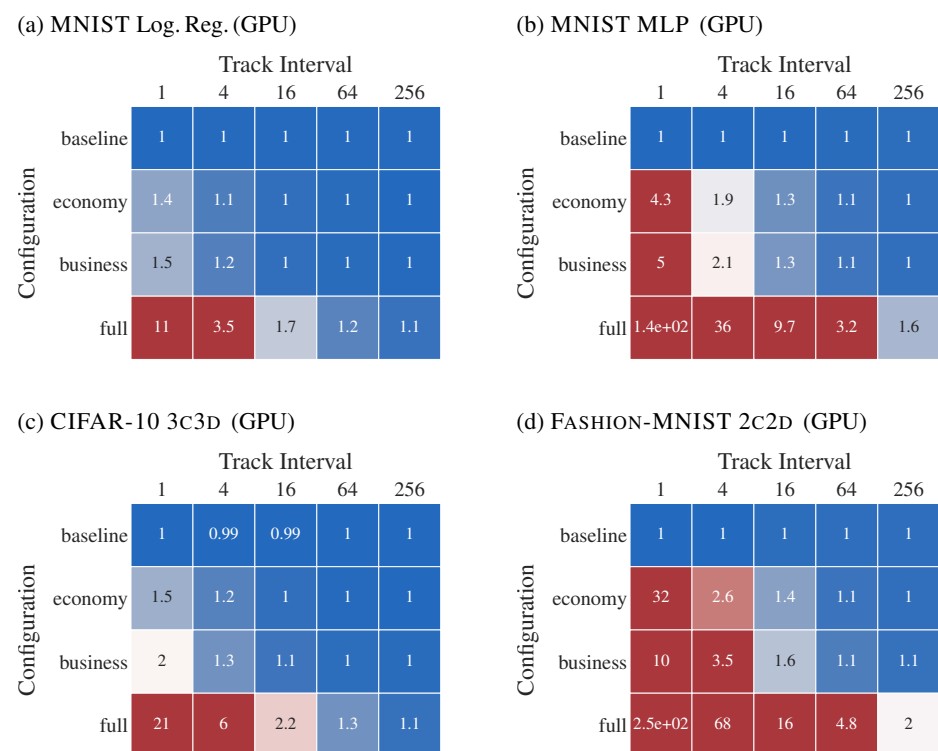

Figure 15: **Overhead of COCKPIT configurations on GPU for four different problems with varying tracking interval.** Color bar is the same as in Figure 6.

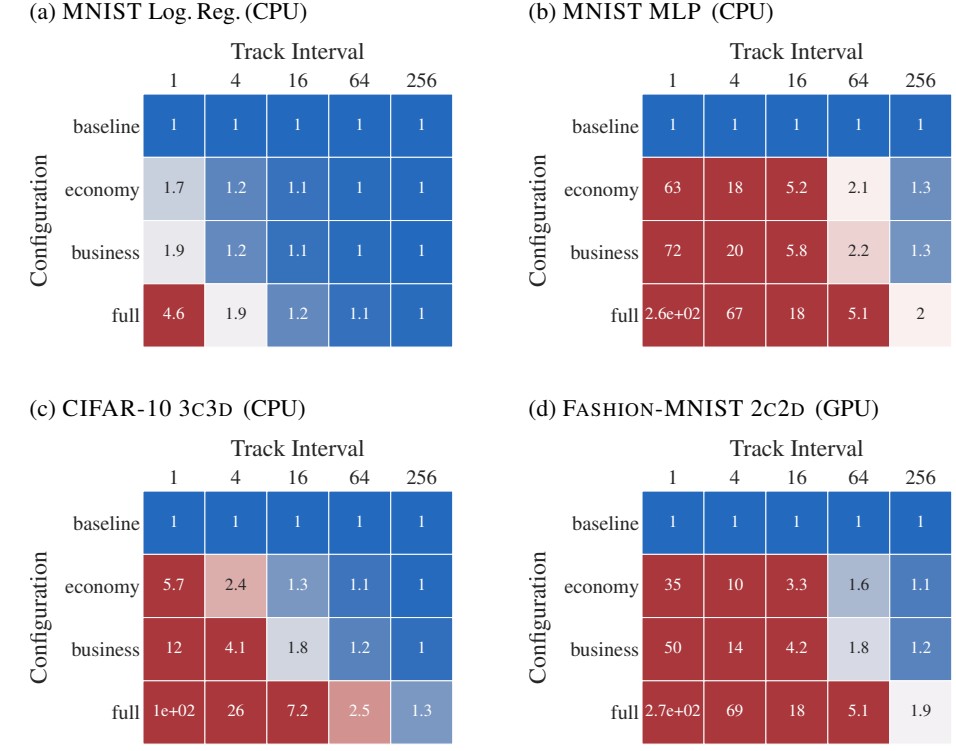

Figure 16: **Overhead of COCKPIT configurations on CPU for four different problems with varying tracking interval.** Color bar is the same as in Figure 6.

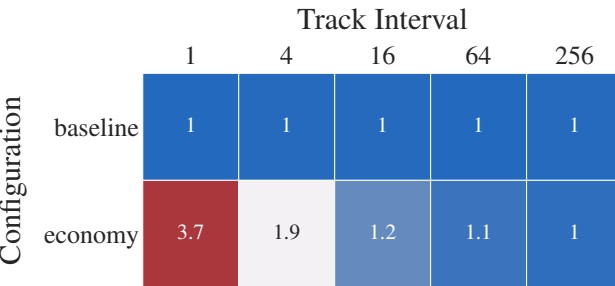

Figure 17: **Overhead of COCKPIT configurations on GPU for RESNET-50 on IMAGENET.** COCKPIT's instruments scale efficiently even to very large problems (here: 1000 classes, $(3, 224, 224)$-sized inputs, and a batch size of 64. For individual gradients to be defined, we replaced the batch norm layers of the RESNET-50 model with identities.) Color bar is the same as in Figure 6.

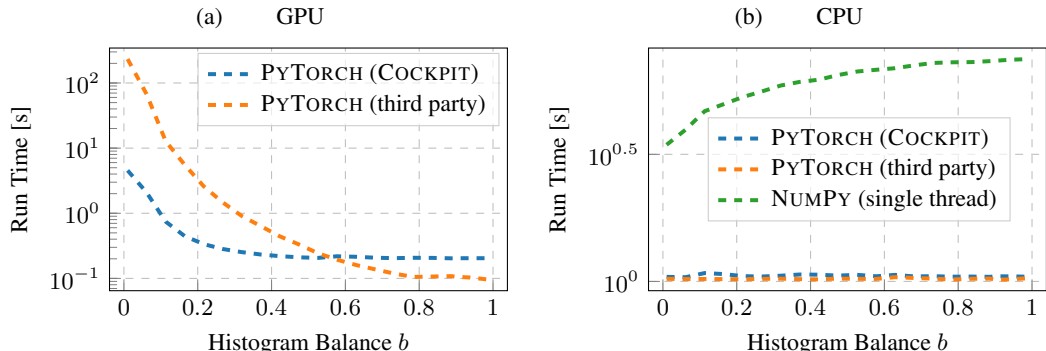

Figure 18: **Performance of two-dimensional histogram GPU implementations depends on the data.** (a) Run time for two different GPU implementations with histograms of different imbalance. COCKPIT's implementation outperforms the third party solution by more than one order of magnitude in the deep learning regime ($b \ll 1$). (b) On CPU, performance is robust to histogram balance. The run time difference between NUMPY and PYTORCH is due to multi-threading. Data has the same size as DEEPOBS's CIFAR-10 3C3D problem ($D = 895,210, |\mathcal{B}| = 128$). Curves represent averages over 10 independent runs. Error bars are omitted to improve legibility.

### E.3 Performance of two-dimensional histograms:

Both one- and two-dimensional histograms require $|\mathcal{B}| \times D$ elements be accessed, and hence perform similarly. However, we observed different behavior on GPU and decided to omit the two-dimensional histogram's run time in the main text. As explained here, this performance lack is not fundamental, but a shortcoming of the GPU implementation. PYTORCH provides built-in functionality for computing one-dimensional histograms at the time of writing, but is not yet featuring multi-dimensional histograms. We experimented with three implementations:

- PYTORCH **(third party):** A third party implementation[7] under review for being integrated into PYTORCH[8]. It relies on `torch.bincount`, which uses `atomicAdds` that represent a bottleneck for histograms where most counts are contained in one bin.[9] This occurs often for over-parameterized deep models, as most of the gradient elements are zero.

- PYTORCH **(COCKPIT):** Our implementation uses a suggested workaround, computes bin indices and scatters the counts into their associated bins with `torch.Tensor.put_`. This circumvents `atomicAdds`, but has poor memory locality.

- NUMPY: The single-threaded `numpy.histogram2d` serves as baseline, but does not run on GPUs.

To demonstrate the strong performance dependence on the data, we generate data from a uniform distribution over $[0, b] \times [0, b]$, where $b \in (0, 1)$ parametrizes the histogram's balance, and compute two-dimensional histograms on $[0, 1] \times [0, 1]$. Figure 18 (a) shows a clear increase in run time of both GPU implementations for more imbalanced histograms. Note that even though our implementation outperforms the third party by more than one order of magnitude in the deep neural network regime ($b \ll 1$), it is still considerably slower than a one-dimensional histogram (see Figure 13 (c)), and even slower on GPU than on CPU (Figure 18 (b)). As expected, the CPU implementations do not significantly depend on the data (Figure 18 (b)). The performance difference between PYTORCH and NUMPY is likely due to multi-threading versus single-threading.

Although a carefully engineered histogram GPU implementation is currently not available, we think it will reduce the computational overhead to that of a one-dimensional histogram in future releases.

---

[7]Permission granted by the authors of `github.com/miranov25/.../histogramdd_pytorch.py`.
[8]See `https://github.com/pytorch/pytorch/pull/44485`.
[9]See `https://discuss.pytorch.org/t/torch-bincount-1000x-slower-on-cuda/42654`

# F  COCKPIT view of convex stochastic problems

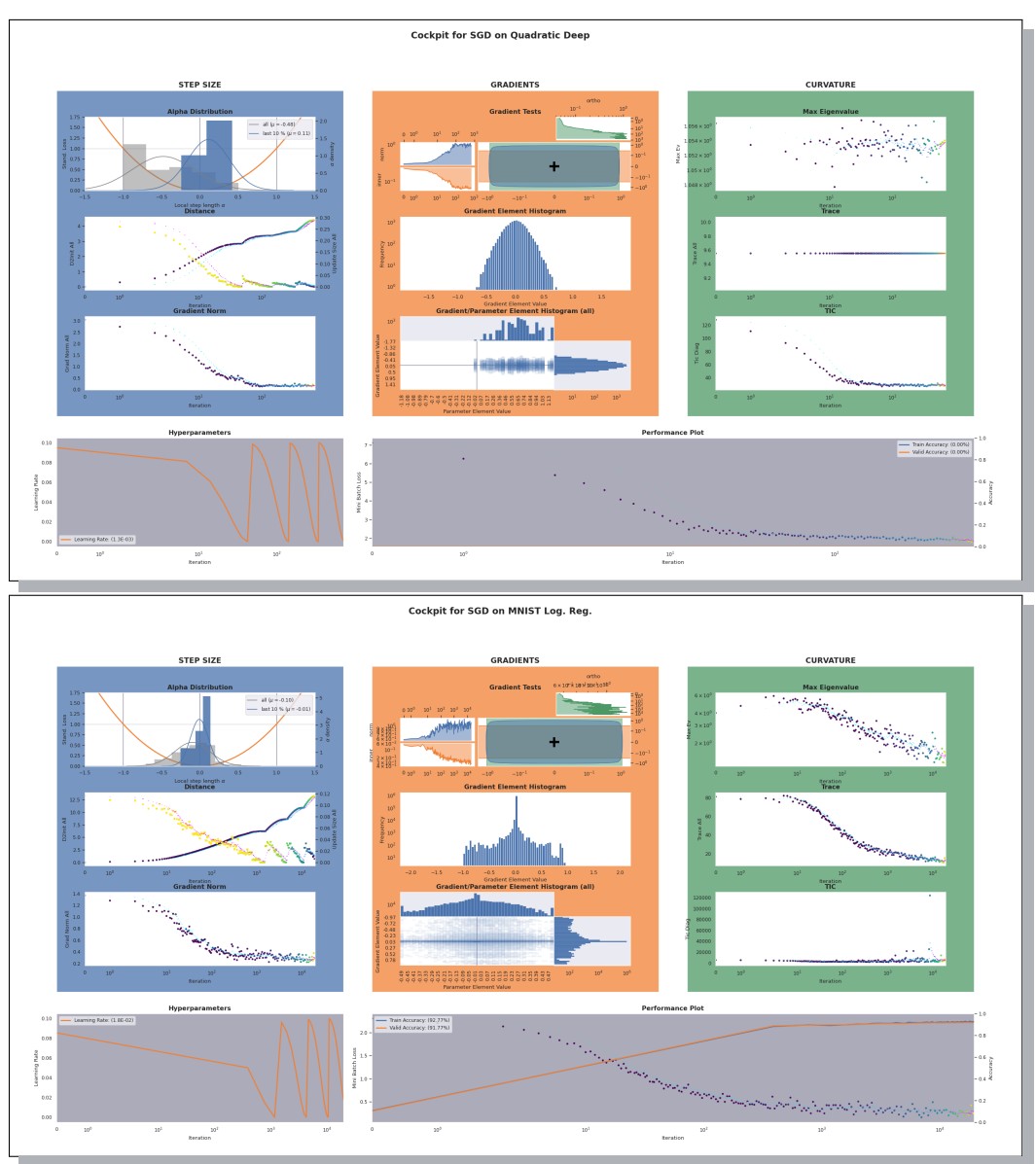

Figure 19: **Screenshot of COCKPIT's full view for convex** DEEPOBS **problems.** Top COCKPIT shows training on a noisy quadratic loss function. Bottom shows training on logistic regression on MNIST. Figure and labels are not meant to be legible. It is evident, that there is a fundamental difference in the optimization process, compared to training deep networks, i.e. Figure 2. This is, for example, visible when comparing the gradient norms, which converge to zero for convex problems but not for deep learning.