# OpenReview forum: "Cockpit: A Practical Debugging Tool for the Training of Deep Neural Networks"
_NeurIPS.cc/2021/Conference — NeurIPS 2021 Poster_

### Official Review · Reviewer_6WN2 · 2021-07-12

**Rating:** 7
**Confidence:** 4

**Summary:**

This manuscript describes an open-source tool for providing useful training-related metrics and insights for deep learning models called Cockpit. Unlike current mainstay deep learning training visualization tools such as TensorBoard, Cockpit provides more in-depth metrics about the model's training state, such as a normalized weight update size, gradient norm, and eigenvalues of hessian. The authors provides motivations for computing and visualizing these metrics, making a case for their usefulness in diagnosing problematic model training. In addition, the authors show that these metrics can be computed with relatively low overhead.

**Limitations And Societal Impact:**

Limitations
1. As stated above, the metrics described in Table 1 and Section 2 includes a wide range of basic and advanced metrics, but Section 3 provides practical use cases only for the more basic metrics. Providing examples of how the more advanced metrics may be helpful to model debugging will strengthen the entire paper significantly. This is also important considering that the more advanced metrics such as HessMaxEv and TICDiag are more expensive to compute and hence require justification for their usefulness.
2. I would like to see more description / discussion of the extensibility of the tool. Due to the rapid pace of the AI/DL field, it's likely that new metrics / quantities will become useful for model training debugging. Hence it is important for the tool to have an extensible API. Is this already the case for Cockpit? If so, what types of information are accessible via the Cockpit API (in combination with the underlying PyTorch library)? Giving some examples would be good.
3. The word "debugging" has overloaded meanings in the field of deep learning. it sometimes refer to troubleshooting low-level bugs such as incorrect implementation of the underlying mathematical formulae / model structure / loss function; it sometimes refer to understanding "why the model is not training properly", which has more to do with hyperparameter choices; it may also refer to understanding why a trained model makes wrong predictions on given examples. This manuscript is mainly about the 2nd type. The authors might want to clarify that.
4. Figure 2 contains multiple subplots that are too small to read and appreciate. The authors refer to different subplots of Figure 2 in various places of Section 2. Unfortunately the English used to refer to the subplots is hard to follow. I suggest the authors use the words "middle" and "center" consistently. "middle" is with regard to the vertical dimension, "center" is with regard to the horizontal dimension. The vertical dimension should always precede the horizontal dimension (e.g., "bottom left", "middle center", "middle right", "top center"). In addition, I suggest adding a column to Table 1 called "location in Figure 2", so it'll be easier for readers to cross reference Table 1 with the figure.
5. Page 3, 1st paragraph. It is unclear why the authors contrasts Cockpit with TensorBoard, and they describe the latter as "just a collection of plots". The primary GUI of Cockpit, as shown in Figure 2, is indeed a collection of plots, albeit containing both scalar curves, histograms and other types of data. TensorBoard as a GUI can also visualize these different types of graphs (see the Scalars, Histograms, and Distributions dashboard of TensorBoard). TensorBoard/TensorFlow doesn't have out-of-the-box support for the list of quantities shown in Table 1, which is the main distinguishing factor of this tool. The authors might want to clarify that.
6. I can't find a place in the paper where optimizers other than SGD is talked about. Given other optimizers (e.g., ADAM, RMSProp) are also widely used, I feel the author should briefly talk about how the use of the more complex optimizers may affect the usage of Cockpit.
6.

**Main Review:**

Originality
- Cockpit goes beyond traditional deep learning monitoring and visualization tools by providing a way to get a suite of 1st-order (gradient) and 2nd-order (hessian) metrics related to a training neural network's weights. There are tools such as TensorBoard and SageMaker Debugger which provide training-related metrics, but none that I know provides metrics as powerful or as up-to-date as the list shown in Table 1 of the paper. I believe the paradigm of this paper will be a valuable addition to the field of deep learning model tooling.

Quality
- Overall, this paper is properly structured, well written, and easy to follow. It provides not only the theoretical basis for the metrics included in Cockpit, but also several practical use cases.
- There is the weakness that the practical use cases (Section 3) cover only a subset of the metrics listed in Table 1 and the subset includes only the more basic metrics such as the gradient histogram and alpha (normalized step size). It would strengthen the paper to provide practical use cases for the more advanced metrics such as HessMaxEv and TICDiag.

Significance
- Debugging model training issues is a common problem experienced by ML / neural network researchers and engineers. Having a more powerful tool to give insights to the reasons for problematic model training and their remedies will shorten the research/development cycle and help improve model quality.

**Time Spent Reviewing:**

2

---

> ### Author Response · Authors · 2021-08-10
> **Reply to Review by Reviewer 6WN2**
>
> Dear Reviewer (6WN2),
>
> thank you for your positive feedback. We are excited you agree that the
> introduced tool will be a valuable addition to the deep learning toolbox. We
> hope our response below alleviates many of your raised concerns and strengthens
> your opinion about the paper. **Please let us know if we can clarify things
> further. We’d be happy to discuss.**
>
> ---
>
> 1. **Utility of more advanced metrics:** Given the page limit, we had to either
>    restrict the number or detail of presented quantities in the main text. We
>    decided to withhold quantities inspired and discussed more extensively by
>    previous works (`HessMaxEV`, `TICDiag`, ..., see Table 1) and only show they
>    can be accessed conveniently and efficiently through Cockpit. In the
>    experiments, we focused on novel metrics (e.g. `Alpha` and layer-wise
>    `GradHist1d`) that have not been evaluated before.
>
>    We understand your concern and want to complement it with a concrete example:
>    Detecting regularization phases during training. E.g. [1] uses
>    $\lambda_{\text{max}}$ as a flatness indicator to show that SGD converges
>    towards the flattest minimum (see Eq. 5 and Fig. 1b in [1]): For the studied
>    problem, SGD reaches areas of small loss early (*optimization phase*), but
>    keeps following the valley towards the flattest minimum (*regularization
>    phase*). The latter phase can be identified through $\lambda_{\text{max}}$,
>    but not by the loss. Monitoring $\lambda_{\text{max}}$ as a flatness
>    indicator for different optimizers, e.g. as done in Fig. 7 of [1], can thus
>    hint at phases where the optimizer still progresses, although the loss
>    stagnates.
>
>    We offer to provide Cockpit's view of the scenario studied in [1] in the
>    appendix.
>
> 2. **Code extensibility:** The code is designed to be easily extensible with new
>    quantities or plots. This is done through fully-documented abstract base
>    classes (see the submitted
>    [code](https://github.com/h1j9d3q/cockpit/blob/master/cockpit/quantities/quantity.py#L9-L25)).
>    We also maintain an anonymized documentation [here](https://h1j9d3q.github.io/cockpit/)
>    that will be updated as we continue the package development to establish
>    Cockpit as a community tool.
>
>    There are no major restrictions to the type of information available through
>    Cockpit: New quantities can rely on the BackPACK library, or on PyTorch's
>    automatic differentiation, e.g. for higher-order differentiation. The most
>    severe design choice in Cockpit is that we do not allow re-evaluating the
>    loss on a new mini-batch (Sec. 2). We think this increases Cockpit's utility
>    as it maximizes computations shared with the actual training loop.
>
> 3. **Clarification of 'debugging' in the deep learning context:** We are indeed
>    mainly discussing the second type of bug. We now added a paragraph in the
>    paper to distinguish three types of bugs in neural networks:
>
>    - *Implementation bugs*, e.g. low-level bugs that could also result in
>      `SyntaxError`s
>
>    - *Training bugs,* e.g. mistakes that result in inefficient or unsuccessful
>      training
>
>    - *Prediction bugs*, e.g. wrong predictions of a trained model on specific
>      examples
>
>    Traditional debuggers are well-suited to find *implementation bugs*. We argue
>    that new tools like Cockpit are necessary to identify *training bugs*.
>
> 4. **Referencing subplots** Good point, thanks! We added the suggested
>    cross-reference overview in Table 1 and now use a consistent way to refer to
>    these subplots by using the column colors in Figure 2.
>
> 5. **Contrast to TensorBoard:** You are right that the difference between
>    TensorBoard and Cockpit is the out-of-the-box support for the carefully
>    selected quantities and making them efficiently available. We clarified our
>    original statement which could indeed be misunderstood.
>
> 6. **Support for other optimizers:** Cockpit's quantities work for general
>    optimizers and can be used identically without increased costs.
>
>    One (current) exception that we will fix for the manuscript's publication is
>    the computation of `Alpha`: It can be computed more efficiently when given
>    access to the optimizer's update step during a backward pass (before the
>    optimizer updates the parameters). This is currently only implemented for
>    SGD, but can easily be done for other optimizers. For now,  if necessary,
>    we automatically fallback to a less efficient but fully general computation
>    of `Alpha` that works for any optimizer.
>
> ---
>
> **References:**
>
> [1] Mulayoff, R., & Michaeli, T. (2020). Unique properties of flat minima in
>   deep networks. In: International Conference on Machine Learning (pp.
>   7108–7118).

---

> > ### Comment · Reviewer_6WN2 · 2021-08-18
> > **Response to author response**
> >
> > 1. Adding the example about regularization detection sounds good to me.
> > 2. Thanks for clarifying the extensibility of the code.
> > 3. This categorization looks right, except that implementation bugs can be higher-level and subtler than Python SyntaxErrors. For example, failure to add an epsilon to a denominator in a division operation used in a neural network can lead to sporadic silent division-by-zero (infinity) problems. Are such problems within the scope of Cockpit?
> > 4 - 6. These all sound good to me.

---

> > > ### Comment · Reviewer_6WN2 · 2021-09-10
> > > **Reply**
> > >
> > > I maintain my original assessment of this manuscript. I expect the authors to address the comments from the reviewers including myself in the camera-ready version of the paper.

---

### Official Review · Reviewer_PQdu · 2021-07-14

**Rating:** 6
**Confidence:** 2

**Summary:**

Authors propose a tool for monitoring and debugging the training process of deep learning models. The tool visualizes/graphs a list of training related mathematical quantities that may improve understanding of the model, its hyperparameters and its training behavior.

**Limitations And Societal Impact:**

I think that authors address the limitations of the tool as much as they can without formal user study. The limitations would be more apparent with actual user study.
I agree with authors that their tool would likely to have positive societal impact.

**Main Review:**

Authors present a collection of tools for visualizing, monitoring and debugging training process of deep neural networks.

I believe that contribution in this area can be very valuable.

However, this paper does not really evaluate the toolkit authors propose. The paper does not evaluate different tools in the toolkit and their contribution to the monitoring and debugging process.

It is possible that Cockpit is a great toolset. However, this cannot be ascertained from the paper.
The illustrative examples shown in sections 3.1 and 3.2 provide some motivation to use the tool, but they do not evaluate user experience.
I realize that tool(kit) evaluation is (very) hard. Running user studies to evaluate the tools takes time and resources. The situation is further complicated when the tool can be used by limited audience of deep learning practitioners.
However, lack of evaluation with actual users makes it harder to know the benefits and drawbacks of the toolkit and its component views.

I have limited experience in training and debugging deep learning models and therefore I cannot evaluate if Cockpit is valuable and which tools in Cockpit are useful.

Changed rating based on the authors' response and discussion.

**Time Spent Reviewing:**

3

---

> ### Author Response · Authors · 2021-08-10
> **Reply to Review by Reviewer PQdu**
>
> Dear Reviewer (PQdu),
>
> thank you for your time reviewing our work and your provided feedback.
>
> We agree that user experience is an important part of any software tool. We
> addressed this in Cockpit's design by making the package easy to set up (see the
> [documentation](https://h1j9d3q.github.io/cockpit/)) and extend. Surely, a user
> study would further highlight Cockpit's utility. However, at this point, we
> believe that it is first important to identify useful observables, from a
> mathematical/mechanistic perspective, before it even becomes possible to study
> how to best present them to the user. A user study should come at a later stage,
> and the community now first has to agree that such tools are needed in the first
> place. This is what we are arguing for with this paper (and we are glad to see
> that some of the other reviewers agree with us on this point).
>
> To argue, why a new type of debugger is necessary at all, we focus on two
> crucial factors that influence the user experience of a debugger:
> Firstly, it must provide meaningful insights, and secondly, come at a reasonable
> computational overhead. We try to showcase Cockpit's benefits with respect to
> both aspects in our paper with concrete examples (Section 3) and objective run
> time numbers (Section 4):
>
> - The experiments aim to show that Cockpit's instruments reveal valuable
>   information that would otherwise remain hidden if only the standard quantities
>   (such as training loss) were inspected. To showcase this, we deliberately
>   inject bugs into well-understood settings and exposed them using our tool. Of
>   course, such a collection is subjective and incomplete. But we believe these
>   bugs not only motivate the tool, but represent relevant failure cases that
>   matter in practice.
>
> - In contrast to the subjective use cases, the objective performance aspect is
>   carefully probed in our Benchmark Section 4 (and Appendix D). There, we assert
>   that Cockpit can be used at reasonable run time overhead.
>
> In our paper, we could further clarify how we addressed the user experience of
> our tool in its design and that further practical experience with the tool (for
> example in form of a large user study) could provide more evidence for its
> utility.

---

> > ### Comment · Reviewer_PQdu · 2021-08-13
> > **Thank you for your comments**
> >
> > Thanks to authors for your comments.
> > I am aware that doing user tool research and getting it accepted at conferences is tough work. Thanks for doing this work!
> > I hope that my comments are helpful.
> > Let me try to split the comments and discussion into discrete areas.
> > - Need for the tool: I think you do a reasonable job in showing the need considering the space limitations in the paper. Ideally, there would be a paper with models/user studies showing just the need for specific tool/features/observables, but that's usually unrealistic.
> > - You put a lot of observables into the tool. Some of them are better motivated than others. It is still unclear to me why a particular collection of observables was chosen and how they fit into a coherent whole rather than being observables that are put into the tool just because you could and/or just because one of them was useful in certain case. This is a difficult and quite ignored area of user tool research - and where user studies should help (but would also take a lot of time and effort).
> > - The observables should be computationally feasible and not too expensive. You do a good job evaluating and presenting this.
> >
> > I disagree with your comment:
> > "at this point, we believe that it is first important to identify useful observables, from a mathematical/mechanistic perspective, before it even becomes possible to study how to best present them to the user"
> > Your paper contradicts your comment: you are presenting the observables to the user via the tool you implemented and published. And this tool is a major contribution of your work. So you have the responsibility for the presentation side of the tool too.
> >
> > With all that said, I want to repeat that I appreciate your work in rather difficult area to publish. I am fine if the conference accepts your paper. Good luck.

---

### Official Review · Reviewer_68Eo · 2021-07-16

**Rating:** 7
**Confidence:** 3

**Summary:**

The core contribution of Cockpit is a piece of software that provides 12 quantities meant to help practitioners debug issues with their model's training. Some of these quantities are novel (to my knowledge), while many of them are existing metrics taken from the literature (although often with improved computational methods or a slightly different setting).

They provide explanations for each quantity and how it's meant to be used, as well as some case studies on how these quantities might help a practitioner debug their model.

Finally, they provide benchmarks demonstrating that the computational overhead of computing these quantities is not overly burdensome on the model.

**Ethical Concerns:**

No ethical concerns.

**Limitations And Societal Impact:**

Yes.

**Main Review:**

Overall, I like this paper a lot. I think it's tackling a very important (and mostly unaddressed!) issue with neural network training - how do we figure out why our model isn't performing well? I would argue that tools like Tensorboard and the metrics they report (e.g. gradient norm) have had a significant impact, and I'd hope that this tool could extend that even further.

In addition, the paper is fairly convincing that the quantities they've chosen may be useful to practitioners, and this is backed up by both the descriptions of the quantities as well as the case studies in the experiments section. The appendix also makes it clear that determining the computation of these quantities was done carefully (although much of the heavy lifting here seems to be done by the underlying package BackPack).

I'd particularly like to highlight the "alpha" quantitity, which is meant to measure whether the gradient is undershooting or overshooting. AFAIK, measuring this quantity in their manner is a novel contribution, and they provide some experiments + intuition for how this quantity can be used to troubleshoot your network's learning rate.

In addition, I appreciate the polished software package (with documentation + examples), as well as close integration with a popular machine learning framework (PyTorch).

My primary criticism is that I wish these quantities were explored more thoroughly. Most of the examples are done on relatively toy datasets (CIFAR-10, SVHN, F-MNIST, CIFAR10). However, considering the benchmarks in this paper, it seems that it would be feasible to explore these quantities in a more realistic setting. In particular, the observation that the "alpha" quantity is >0 seems quite interesting to me, I would like to see it explored more.

Some nitpicks/minor questions
- There are these ... random thin black lines around some of the figures, such as near Figure 1. I assume they aren't intentional.
- I think both the paper/appendix as well as the documentation would benefit from an explanation of how each quantity may be commonly used to debug training.


**Time Spent Reviewing:**

5

---

> ### Author Response · Authors · 2021-08-10
> **Reply to Review by Reviewer 68Eo**
>
> Dear Reviewer (68Eo),
>
> many thanks for your feedback. We are glad you like the paper and find it
> valuable! We hope you will continue to argue in our favor during the
> discussion. Regarding your minor points:
>
> - Thank you for notifying us about the random black lines. They did not show up
>   in any of the PDF viewers we used. We have since identified that they seem to
>   occur only on Apple products. We will fix this for the camera-ready version.
> - We added a statement of how each quantity may be used to debug training in its
>   description.

---

> > ### Comment · Reviewer_68Eo · 2021-08-18
> > **Thanks for Comments**
> >
> > After reading the other reviewer's comments and response, I stand by my score. However, I do want to highlight a couple points other reviewers made that I agree with.
> >
> > 1. As I (and other reviewers) have mentioned, it is suboptimal that all of the experiments were done on fairly toy datasets. There are a multitude of issues that could potentially arise from larger models, from differences in training dynamics to computational feasibility. Simply showing some experiments on ImageNet with larger models (like you suggested in your response) would do a long way to alleviating these concerns.
> >
> > 2. As one of the other reviewer mentioned, it's a weakness that the practical use cases cover only a subset of provided metrics. Providing a couple more case studies would go a long way to making this paper more convincing.
> >
> > I agree with other reviewers that an user study would be quite nice, even on quite a toy scale. For example, you could take one of the examples with suboptimal training and give it to a couple different researchers (say 5 to 10). Even such a toy user study would be convincing imo. I definitely don't think that this paper needs an user study to be accepted though.

---

> > > ### Comment · Reviewer_PQdu · 2021-08-18
> > > **Good comments**
> > >
> > > I agree with your comments.

---

### Official Review · Reviewer_EHzW · 2021-07-17

**Rating:** 5
**Confidence:** 4

**Summary:**

The paper proposes to visualize 12 different metrics derived from the neural network training process, with the intention to help deep learning practitioners debug issues encountered during neural network training. These metrics are derived from the loss, the parameters, and first/second-order information from the parameters' gradients. The authors demonstrate the effects of incorrectly scaled training data, vanishing gradients due to use of a suboptimal activation function, and various learning rates on these metrics. The authors show the run-time performance of computing the metrics, and conclude with a demonstration of the metrics on a successful training run of All-CNN-C on CIFAR-100.

**Limitations And Societal Impact:**

I believe the authors have been straightforward about what the work can do, including limitations.

I would suggest to the authors that they gather more experience from a variety of users who can use the Cockpit tool to debug real-world issues with their neural network training. Alternatively, the authors can try to gather common bugs found in neural architecture design and training and reproduce them using the tool, to determine tell-tale signs for these common bugs inside Cockpit. The authors can then automatically provide suggestions for what may be going wrong inside the tool, or produce a guidebook or manual for how to debug using it.

A more formal user study, even small in scale, to show that users can benefit from using the tool, would also be quite helpful.

**Main Review:**

## Originality
The idea of visualizing the training progress of a neural network is not new, and tools like TensorBoard allow building dashboards to report various kinds of progress. Many of the metrics used in the work also come from existing work (as cited in Table 1). Nevertheless, the software in this work conveniently allows the users to utilize these metrics in one place. The paper also provides some suggestions for how these disparate metrics might be interpreted together to get a fuller picture.

## Quality
The authors show that useful information can be derived from the metrics proposed in the paper. The set of metrics seems reasonably well thought-out and the software package appears complete. However, I found the following lacking about the experiments and the analysis of the work:
- The neural architectures and datasets used to evaluate the work are all fairly small, consisting of MLPs or CNNs with a handful of layers. The datasets used are CIFAR-10, SVHN, MNIST, and Fashion-MNIST. For these smaller models and datasets, it is already fairly well-understood how to train them effectively. Since they are small, it is also much cheaper to use black-box hyperparameter optimization methods rather than trying to debug issues like a suboptimal learning rate. The work would be improved if it evaluated using larger datasets and more complex architectures and objectives, especially those that are considered more difficult to train like GANs or Transformers.
- The authors propose that the work can tell the user why neural network training is failing. Indeed, the authors show the effect that incorrectly scaled data and the use of a suboptimal activation function can have on the metrics computed by the provided tool. However, it is not clear how the user should be able to determine the underlying cause from what is shown by the tool.
  - For incorrectly scaled inputs leading to a different gradient distribution, it is not clear that the changed gradient distribution has to be bad. There are also other potential causes, like incorrectly scaled initial weights.
  - For vanishing gradients, there are other potential causes like not using batch normalization which could cause the same issue. Also it seems plausible that some patterns for layer-wise histograms may seem intuitively bad but nevertheless turn out to work well in practice.
- Similarly, the authors suggest using a standardized step size to determine whether the learning rate is too small or too big. However, Figure 5 shows that certain step sizes are good for certain problems but bad for others, making it a difficult measure to use when provided with a new problem.

## Clarity
The paper is clearly written and straightforward to understand. It may make more sense to put the Benchmark section after the Showcase section, since the Showcase section references a figure near the start of the paper and also is more related to the Experiments section.

## Significance
I think the tool itself can be useful for deep learning practitioners, but I don't believe that this paper clearly demonstrated exactly in which ways the tool would be useful.

**Time Spent Reviewing:**

3

---

> ### Author Response · Authors · 2021-08-10
> **Reply to Review by Reviewer EHzW**
>
> Dear Reviewer (EHzW),
>
> thank you for your thorough and constructive feedback. We will gladly swap the
> Benchmark and Showcase sections to improve our paper's clarity:
>
> 1. **Experimental models are small and well-understood:** We agree with you that
>    Cockpit's largest benefits apply to training novel models on non-standard
>    data sets. For our experiments, however, we deliberately picked models and
>    data sets *because* they are well-understood. Artificially introducing bugs
>    into these widely-known examples allows us to showcase Cockpit's capabilities
>    in the most transparent way possible on problems that every user knows and
>    understands. As we show in Cockpit's performance evaluation, it scales to
>    larger architectures. To account for this, we'd be happy to (for example)
>    change our experiment with incorrectly scaled data to a larger data set such
>    as `ImageNet`. Would this convince you of Cockpit's scalability?
> 2. **Bugs could have different sources:** Yes, the presented bugs could
>    potentially also have other sources. This is a natural consequence of the
>    vast complexity of model training and neural networks. But this also shows
>    how important a tool like Cockpit can be to narrow down these bugs instead of
>    black-box probing the training procedure. Without such a tool, the
>    practitioner has virtually no information on which part of the training
>    pipeline causes the bug. Our tool is able to reduce the list of potential
>    sources down to a few possibilities.
>    Unfortunately, there are no simple rules to identify and solve complex bugs;
>    otherwise, this process would have already been automated. But by inspecting
>    the internals with Cockpit one can form hypotheses about what went wrong and
>    specifically probe them instead of using plain trial-and-error.
> 3. **Unclear what an optimal normalized step size would be:** Our experiment
>    that showcases the `Alpha` quantity is meant to highlight two aspects:
>    - *Besides identifying bugs, our tool can also help with tuning.* Admittedly,
>      this does not mean that it can automatically detect the optimal learning
>      rate for any given problem. This would be a significant contribution in
>      itself as it would provide a tuning-free optimization method. Instead, the
>      `Alpha` quantity quickly identifies a large set of learning rates as
>      sub-optimal (e.g. learning rates that lead to negative `Alpha`).
>    - *More importantly, our presented tool can even help with research.* Perhaps
>      this is not the most obvious use case of a debugging tool. But looking at
>      neural network training through the lens of Cockpit's `Alpha` quantity
>      identifies a need for *"overshooting"* or *"surfing the wall"*.
>      This motivates new research into tuning-free methods that follow such a
>      policy instead of locally minimizing the loss (as done by most current
>      methods).
> 4. **Provide a manual or guidebook:** You suggested to "gather common bugs [...]
>    and reproduce them using the tool". This is essentially what we were aiming
>    at in Section 3 by studying bugs caused by data (3.1), model (3.2), and
>    hyperparameters (3.3)! Of course, a 9 page paper cannot possibly list all
>    common bugs in deep learning models. *Which particular bugs would you have
>    liked to see?* We want to kick off an endeavor to improve the current state
>    of monitoring training and debugging deep learning. Providing an extensive
>    manual with a clear "if-X-then-Y" instruction would more or less "solve deep
>    learning" and is therefore beyond the scope of this work: It will require the
>    joint community effort. By efficiently providing numerous debugging
>    quantities, we believe Cockpit can be an important puzzle piece in this
>    endeavor.
>
> Let us know in a reply if we can clarify things further. We are happy to
> discuss. If our clarifications were able to alleviate some of your concerns
> please consider revising your score.

---

> > ### Comment · Reviewer_EHzW · 2021-08-24
> > **Response to reply**
> >
> > Thank you to the authors for the well-reasoned response.
> >
> > I agree with your overall comment that it is too much to expect a single paper like this to "solve deep learning", and that should not be a reason to reject the paper. As you pointed out in your paper and your response, the various metrics presented in the tool can help developers and researchers find issues with their models, even if the tool cannot directly point out exactly what is wrong.
> >
> > I am fine with the paper being accepted, but I think there is also some consensus among the reviewers that 1) a user study where people use the tool to debug a problem, and 2) studies on larger models, would be useful. I hope that the authors consider these directions for improvement in any future versions of the paper.

---

> > > ### Author Response · Authors · 2021-09-01
> > > **Request for larger models**
> > >
> > > Dear Reviewer (EHzW),
> > >
> > > thanks for your thoughtful comments.
> > >
> > > To account for your concerns regarding studies on larger models, we would adapt
> > > the experiment with incorrectly scaled data to `ImageNet`, as offered in our
> > > original reply. Cockpit's performance evaluation already includes such scenarios
> > > (see Figure 15 in the Appendix for `resnet50` on `ImageNet`).
> > >
> > > We hope this addresses your request for studies on larger models.

---

### Decision · Program_Chairs · 2021-09-27

**Decision:**

Accept (Poster)

**Comment:**

The reviews for this paper are quasi-unanimous, in the sense that the one reviewer who voted to reject has said they'd be OK with the paper being accepted.
I do agree with some of the reviewers that experiments on larger models and more case studies would be nice, and part of me wants to recommend that this paper be rejected with the feedback to make those changes and resubmit.
However, another part of me is worried that, with the reviewing process being as noisy as it is, that might result in this paper being rejected in the future - especially since I do think reviewers / ACs tend to have (unfairly so) a higher bar for these types of contributions.
Given that, I will recommend acceptance.
I would really like to see a version of this paper with those changes made, however.